# Multicentric study of cervical cancer screening with human papillomavirus testing and assessment of triage methods in Latin America: the ESTAMPA screening study protocol

Maribel Almonte ,[1] Raúl Murillo,[2] Gloria Inés Sánchez,[3] Paula González,[4] Annabelle Ferrera,[5] María Alejandra Picconi,[6] Carolina Wiesner,[7] Aurelio Cruz-Valdez,[8] Eduardo Lazcano-Ponce,[8] Jose Jerónimo,[9] Catterina Ferreccio,[10] Elena Kasamatsu,[11] Laura Mendoza,[11] Guillermo Rodríguez,[12] Alejandro Calderón,[13] Gino Venegas,[14,15] Verónica Villagra,[16] Silvio Tatti,[17] Laura Fleider,[17] Carolina Terán,[18] Armando Baena,[1] María de la Luz Hernández,[1,19] Mary Luz Rol,[1] Eric Lucas,[20] Sylvaine Barbier,[1] Arianis Tatiana Ramírez,[1] Silvina Arrossi,[21] María Isabel Rodríguez,[11] Emmanuel González,[22] Marcela Celis,[7] Sandra Martínez,[7] Yuly Salgado,[7] Marina Ortega,[23,24] Andrea Verónica Beracochea,[25,26] Natalia Pérez,[27] Margarita Rodríguez de la Peña,[28] María Ramón,[29] Pilar Hernández-Nevarez,[8] Margarita Arboleda-Naranjo,[30] Yessy Cabrera,[5] Brenda Salgado,[31] Laura García,[32] Marco Antonio Retana,[33] María Celeste Colucci,[6] Javier Arias-Stella,[34] Yenny Bellido-Fuentes,[9] María Liz Bobadilla,[16] Gladys Olmedo,[16] Ivone Brito-García,[8] Armando Méndez-Herrera,[8] Lucía Cardinal,[17] Betsy Flores,[18] Jhacquelin Peñaranda,[18] Josefina Martínez-Better,[35] Ana Soilán,[23,36] Jacqueline Figueroa,[37] Benedicta Caserta,[38] Carlos Sosa,[39] Adrián Moreno,[28] Juan Mural,[28] Franco Doimi,[29] Diana Giménez,[40] Hernando Rodríguez,[40] Oscar Lora,[18,41] Silvana Luciani,[42] Nathalie Broutet,[43] Teresa Darragh,[44] Rolando Herrero[1,4]

Dr Paula Gonzalez deceased on 26 March 2020

**Correspondence to**
Dr Maribel Almonte;
almontem@iarc.fr

## ABSTRACT

**Introduction** Human papillomavirus (HPV) testing is replacing cytology in primary screening. Its limited specificity demands using a second (triage) test to better identify women at high-risk of cervical disease. Cytology represents the immediate triage but its low sensitivity might hamper HPV testing sensitivity, particularly in low-income and middle-income countries (LMICs), where cytology performance has been suboptimal. The ESTAMPA (EStudio multicéntrico de TAMizaje y triaje de cáncer de cuello uterino con pruebas del virus del PApiloma humano; Spanish acronym) study will: (1) evaluate the performance of different triage techniques to detect cervical precancer and (2) inform on how to implement HPV-based screening programmes in LMIC.

**Methods and analysis** Women aged 30–64 years are screened with HPV testing and Pap across 12 study centres in Latin America. Screened positives have colposcopy with biopsy and treatment of lesions. Women with no evident disease are recalled 18 months later for another HPV test; those HPV-positive undergo colposcopy with biopsy and treatment as needed.

Biological specimens are collected in different visits for triage testing, which is not used for clinical management. The study outcome is histological high-grade squamous intraepithelial or worse lesions (HSIL+) under the lower anogenital squamous terminology. About 50 000 women will be screened and 500 HSIL+ cases detected (at initial and 18 months screening). Performance measures (sensitivity, specificity and predictive values) of triage techniques to detect HSIL+ will be estimated and compared with adjustment by age and study centre.

**Ethics and dissemination** The study protocol has been approved by the Ethics Committee of the International Agency for Research on Cancer (IARC), of the Pan American Health Organisation (PAHO) and by those in each participating centre. A Data and Safety Monitoring Board (DSMB) has been established to monitor progress of the study, assure participant safety, advice on scientific conduct and analysis and suggest protocol improvements. Study findings will be published in peer-reviewed journals and presented at scientific meetings.

**Trial registration number** NCT01881659

## Strengths and limitation of this study

► The study design in which additional samples from all women screened with HPV and cytology are collected at screening simulating a reflex-testing scheme whenever possible, will allow the evaluation of triage techniques without influencing the outcome of the study, as tests are performed or evaluated after disease confirmation.

► This is the largest cervical screening study in Latin America with more than 500 histologically confirmed high-grade squamous intraepithelial lesions (HSILs) expected that will permit evaluation of several triage tests, alone or in combination, for HPV-positive women.

► A large number of women are being screened with HPV testing in Latin America, where the majority of women would not otherwise benefit from high-quality cervical screening.

► The multicentric nature of the study will allow capturing experiences from areas which are geographically, culturally and socioeconomically distinct from each other and with different health systems/areas that may face common challenges but that require different approaches in accordance with their context.

► Colposcopy and collection of biopsies was not performed in HPV-negative women (only in a subset who had abnormal cytology), potentially introducing verification bias when assessing absolute performance measures of screening tests to be used in primary screening; however, the study design will allow unbiased evaluation of triage tests.

## INTRODUCTION

More than 500 000 new cases and nearly 300 000 deaths of cervical cancer occur every year, more than 90% in low-income and middle-income countries (LMICs).[1 2] Cytology-based screening programmes have reduced cervical cancer in high-income countries (HICs) but, with few exceptions, not in LMICs. Programmes using cervical cytology are complex and the method has limited sensitivity and low reproducibility, imposing the need for repeated tests, resulting in high cost and logistic complications which hamper programme implementation and success.

It is now clear that a group of about 12 human papillomavirus (HPV) types are the causal agents of cervical cancer and that HPV16 and HPV18 are responsible for about 70% of tumours. HPV is a very common infection usually acquired shortly after initiation of sexual activity, but most infections are cleared by the immune system within 2 years of acquisition and only a few persist and progress to cancer.[3]

Highly effective and safe vaccines against HPV16 and HPV18 have been developed[4] and vaccination programmes are being rolled-out around the world, but the full public health impact of the vaccine is expected only after several decades. Cervical cancer screening programmes remain high priority, especially for LMIC and constitute one of the main interventions to achieve elimination of cervical cancer as a public health problem.

Currently, there are several tests for HPV detection with high sensitivity and reproducibility that can detect more cervical disease at an earlier stage and offer long-term reassurance of low risk of cervical precancer and cancer. These tests are now being used or considered to replace cervical cytology in primary screening as they allow extension of the screening interval, with consequent savings that can compensate the possibly higher cost of the test compared with cytology. In addition, HPV testing can be done on self-collected samples, increasing screening uptake.[5] Furthermore, emerging point-of-care tests giving immediate results can improve treatment rates.

Screening with HPV has the problem that transient HPV infections are very common, particularly among young women, where the majority of infectious will regress spontaneously. Even among women over 30 years of age, HPV infection tends to regress and only in a fraction of women with persistent infection, it can lead to true cancer precursors and cervical cancer. Thus, one main issue to resolve is which tests or strategies can better select HPV-positive women, who are most likely to have or develop the significance disease (triage) in the near future, for further evaluation and treatment.

Here we detail the design of a multicentric study that aims to evaluate visual methods, cytology-based and novel molecular-based techniques that can be used to triage women who test positive for HPV and that can lead to the establishment of HPV-based efficient, affordable and sustainable screening programmes.

## METHODS AND ANALYSIS

The aims of the study are: (1) to investigate the performance of emerging cervical cancer screening and triage techniques among women 30 years and older and (2) to evaluate the feasibility of different approaches for implementation of organised HPV-based screening programmes.

The primary objective is to estimate performance characteristics (sensitivity, specificity, positive and negative predictive value) of multiple techniques alone or in combination for detection of histologically confirmed cervical high-grade squamous intraepithelial lesion (HSIL) or worse lesions (HSIL+) under the Lower Anogenital Squamous Terminology (LAST[6]) among HPV-positive women 30–64 years old.

Secondary objectives are: (1) similar performance analyses among all recruited women and restricted to those with negative cytology; (2) estimations of colposcopy referral, overdiagnosis and overtreatment rates by a single technique or a combination of techniques; (3) establishment of a biological specimens' bank with an associated database to evaluate future cervical cancer screening and triage techniques; and (4) assessment of the feasibility of implementing organised HPV-based screening programmes within local health systems. The methodology to assess this objective will be reported separately.

The key hypotheses of the study are:
1. Cervical cancer screening by HPV testing followed by triage with one or more additional tests identifies the majority of women at high risk of having precancerous

cervical lesions who need treatment to prevent cervical cancer.

2. The number of screened women lost to follow-up in the screening process (receiving screening results, attending diagnostic workup, receiving adequate treatment) could be reduced if: (1) women are well informed of the process and trust the healthcare system, (2) healthcare professionals are well trained and ready to offer care and (3) the follow-up is centrally organised with capacity to contact screenees.

3. HPV-based cervical cancer screening could be implemented in Latin America if a screening platform is developed, and affordability and sustainability of the screening programme can be guaranteed ahead of implementation.

The study is being conducted in 12 study centres in Argentina, Colombia, Paraguay, Bolivia, Costa Rica, Honduras, Mexico, Peru and Uruguay.

Recruitment started in May 2013 and will be completed in December 2020. However, as the study includes a follow-up visit after 18 months of initial screening, final completion of the study is envisaged for July 2022.

### Candidate triage techniques for HPV-positive women

Different tests and approaches will be evaluated for triage of HPV-positive women. Some of them may also be evaluated as stand-alone without triage test: one with enough sensitivity to be used for primary screening but good specificity so that no triage is required.

The alternatives for triaging HPV-positive women include visual methods, cytology, high-risk and type-specific HPV persistence, HPV genotyping, HPV oncoproteins and other novel molecular biomarkers.

### Visual methods

Visual inspection of the cervix after the application of 5% acetic acid (VIA) is inexpensive, simple and can be carried out by primary care personnel (nurses, midwives, general doctors). The sensitivity and specificity of VIA are limited and highly dependent on training and experience of examiners, who require continued training and supervision.[7–9] VIA results are highly heterogeneous, expressing lack of consistency in human judgement.[10] The performance of the test decreases with increasing age due to the regression of the transformation zone into the endocervix in women older than 50 years;[11] however, a recent study showed that VIA might have uniform sensitivity across age groups.[12]

VIA may be used as triage of HPV-positive women in remote areas where high-tech methods for diagnosis may not be available, identifying who can be treated with ablative treatment, who should be referred to a higher healthcare level and who can be followed up without treatment.[13] Current WHO guidelines recommend this as one of the possible screening strategies.[14]

In this study, VIA will be standardised to allow evaluation of whether a lesion is present or not and if the observer considers that the subject would be a candidate for immediate ablative treatment, but VIA results will not be used for clinical management except when cancer is suspected and the woman should be immediately referred to colposcopy. Visual inspection after the application of iodine lugol, the other naked-eye visual method, will be evaluated in some study centres under a different protocol.

### Cytology

Conventional cytology or Pap has been used since the 1950s in primary screening and has succeeded in decreasing cervical cancer in several HICs, particularly where women have had cytology frequently and high adherence to diagnosis and treatment has been achieved. This has not happened in LMIC, where usually coverage has been low and access to diagnosis and treatment has been limited.

Liquid-based cytology (LBC) uses the same principles and criteria of Pap and is replacing Pap in HICs because it offers the opportunity for reflex testing and significantly reduces the number of inadequate samples,[15] although in terms of screening performance both are comparable.[16 17] More recently, computer-scanned liquid-based slide algorithms showed promising results by improving detection of HPV-positive cervical intraepithelial neoplasia grade 3 (CIN3)/women with adenocarcinoma in situ (AIS).[18]

HPV testing followed by triage with cytology (Pap or LBC) is the immediate screening modality to be used in HPV-based screening settings with high cytology capacity. However, the limited sensitivity of cytology might detriment HPV testing sensitivity, especially in places where strict quality assurance cannot be ensured.

It has been postulated that prior knowledge of HPV status may improve the performance of cytology since the interpreter will be more meticulous when reading a smear of an HPV-positive woman. However, results from studies have not been fully consistent; some have shown increases in the proportion of abnormal results[19] leading to increased CIN grade 2 or worse lesions (CIN2+) detection[20] and potential reduction of immediate referrals to colposcopy,[21] while at least one study reported losses in cytology accuracy with increases of false-positive readings and losses in specificity.[22] We plan to evaluate the performance of cytology, both Pap and LBC, with and without prior knowledge of HPV status, as triage tests.

### HPV persistence as a triage strategy

It has been proposed to use 1 year HPV persistence (overall or type-specific) to follow HPV-positive/cytology-negative women in places where cotesting is recommended like in USA or after using cytology for triage of HPV-positive women, like in Argentina.[23–25] Within an HPV-based screening programme where cytology will not be available, a repeat HPV test could also be used to define clinical management. We plan to evaluate the performance of HPV persistence as triage of HPV-positive women, both among all HPV positives and restricted to those with negative cytology. For this purpose, we will use

two time HPV testing points: 1–3 months at colposcopy and 18 months after initial screening among untreated women.

## HPV genotyping

Stratifying women on the presence of HPV16 and HPV18, which are responsible for about 70% of cervical cancer and their precursors, will identify women at the highest risk of CIN3 or worse lesions (CIN3+). In a large cohort study in the USA, among women older than 30 years, the 18 year cumulative incidence of CIN3+ among one-time HPV16-positive women was 8.5% compared with 3.1% for other oncogenic HPV-positive women negative for HPV16,[26] indicating that women positive for HPV16 (and possibly HPV18) should be referred to colposcopy immediately while other HPV-positive women could be recalled later maintaining adequate sensitivity.[26]

Some of the HPV screening tests already provide separate results on HPV16 and/or HPV18, and we will evaluate the performance of HPV16 and HPV18 among HPV-positive women who will be screened with the COBAS HPV test (Roche) in some study centres.

Additionally, the potential to improve the accuracy of HPV testing and triage with different combinations of HPV types is under evaluation,[27 28] and we also envision to carry out full HPV genotyping of all women in the study cohort.

## HPV oncoproteins

E6 and E7 oncoproteins are the main effectors of the HPV oncogenic activity. The molecular structure, functions and expression levels are different between oncogenic and non-oncogenic HPV types.[29] E6 is expressed at elevated levels in cervical cells only when HPV-infected cervical cells undergo precancerous or cancerous changes.

An HPV E6 strip test, the OncoE6 Cervical Test targeting E6 HPV16 and HPV18 (previously Advantage HPV E6 test, targeting E6 HPV 16, 18 and 45) has been used in studies around the world with promising results.[30–33] In a study in Honduras, the sensitivity of the test to detect precancer related to HPV16/18 infections was 96.8% (95 % CI 83.8% to 99.8%) and 56.4% (95% CI 43.3% to 68.6%) regardless of HPV type, and the specificity was 97.5% (95% CI 93.7% to 99.0%). All but one subject with histological HSIL tested positive for E6, and the test was negative in all cases associated with HPV types other than 16 and 18 or in HPV negatives.[31] A new E6/E7 prototype adding oncoproteins for HPV types 31, 33, 35, 45, 52 and 58 has been developed with the aim of increasing overall sensitivity of the method. Initial evaluation of its performance to detect CIN3 +has been carried out by our group. Results were promising and the prototype is under refinement for further clinical evaluation.

## Markers of HPV-induced cell regulation alterations

Another important biomarker under intensive study is $p16^{ink4a}$, a cyclin-dependent kinase inhibitor markedly overexpressed in cancerous and precancerous cervical tissue. It corresponds to a cellular correlate of increased expression of the viral E7 oncoprotein that disrupts the retinoblastoma protein (pRb) pathway, leading to compensatory overexpression of $p16^{ink4a}$.[29] The cellular accumulation of $p16^{ink4a}$ can be measured using ELISA and by immunostaining of histology and cytology slides as done by the CINtec test (Roche, before mtm laboratories).

A meta-analysis of the performance of this test,[34] indicated that the proportion of smears overexpressing $p16^{ink4a}$ clearly increased with severity of cytological (12% normal, 89% HSIL) and histological (2% normal, 82% CIN3) abnormalities. The authors noted limited reproducibility due to insufficient standardised interpretation of the immunostaining. One study comparing Pap, HPV testing and HPV triaged with $p16^{ink4a}$ found that the latter scheme maintains the sensitivity gained by the HPV test but with a referral rate to colposcopy similar to that of a Pap-based programme.[35]

Another biomarker of the cell cycle is Ki-67 which is a proliferation marker. As described, E7 leads to accumulation of $p16$ but also commits the cell into proliferation which leads to overexpression of Ki-67. Since overexpression of $p16$ (tumour suppression protein) and Ki-67 (proliferation marker) are mutually exclusive under normal conditions, the detection of overexpression of both simultaneously by dual-immunostaining could identify cells with deregulated cell cycle. This dual-staining test can substantially simplify and standardise the evaluation of stained slides.[36] In a recent study, dual-staining showed better risk stratification, requiring substantially fewer colposcopies for CIN3 +detection compared with Pap, suggesting that it can safely replace Pap as triage of HPV positives.[37]

## Methylation

DNA methylation is a mitotically transmitted epigenetic motif that reflects molecular events in host cells contributing under certain conditions to cervical carcinogenesis. It can be quantitated with good accuracy in exfoliated cells and could be applied directly to the residual sample after HPV test.[38] Several human genes have consistently shown elevated methylation in cervical precancer, highlighting the potential role of methylation for triage of HPV-positive women.[39 40] Additionally, aberrant methylation of certain HPV genes including L1 and L2 are also associated with precancer, especially for types HPV16, HPV18, HPV31, HPV33 and HPV45.[41 42] A recent study showed that increased methylation of L1/L2 sites in all 12 high-risk HPV types was positively associated with CIN3/AIS suggesting the role of methylation in marking the transition from infection with high-risk HPV to precancer. In coming years, it is expected that several HPV DNA methylation assays that could serve as triage for HPV-positive women will be developed.

## Participants

Women are invited to screening using different approaches, such as: door-to-door census of the

recruitment area, invitation by community leaders or media campaigns.

Women eligible for inclusion are those aged 30–64 years and residents of a specified catchment area, mentally competent to understand the consent form, able to communicate with study personnel and physically able to have a pelvic exam. Women who have not initiated sexual life, who have a history of cervical cancer or treatment for cervical precancer within the previous 6 months and those with hysterectomy or serious pre-existing medical conditions or plans to move out of the study area in the next 12 months are excluded. The participation of pregnant women and those who have given birth less than 3 months before screening are deferred until they are 3 months or more postpartum or up to the end of the enrolment period. Similarly, women with heavy vaginal bleeding or severe cervical infection are deferred until the condition is resolved.

## Study design

This is a screening study in which HPV testing and Pap will be carried out in up to 50 000 women aged 30–64 years in multiple centres across Latin America. HPV-positive women and those with abnormal Pap will be referred to colposcopy with biopsy if needed. Identified high-grade cervical lesions will be treated with large loop excision of the transformation zone (LLETZ). Women with no evident disease will be recalled 18 months since initial screening for another HPV test and those positives will undergo colposcopy with biopsy, followed by treatment as needed.

The study is organised in four main clinical visits: Visit 1: Initial Screen, Visit 2: Initial Colposcopy, Visit 3: Follow-up Screen 18 months after the initial screen and Visit 4: Final Colposcopy on HPV positives in the follow-up screen (figure 1).

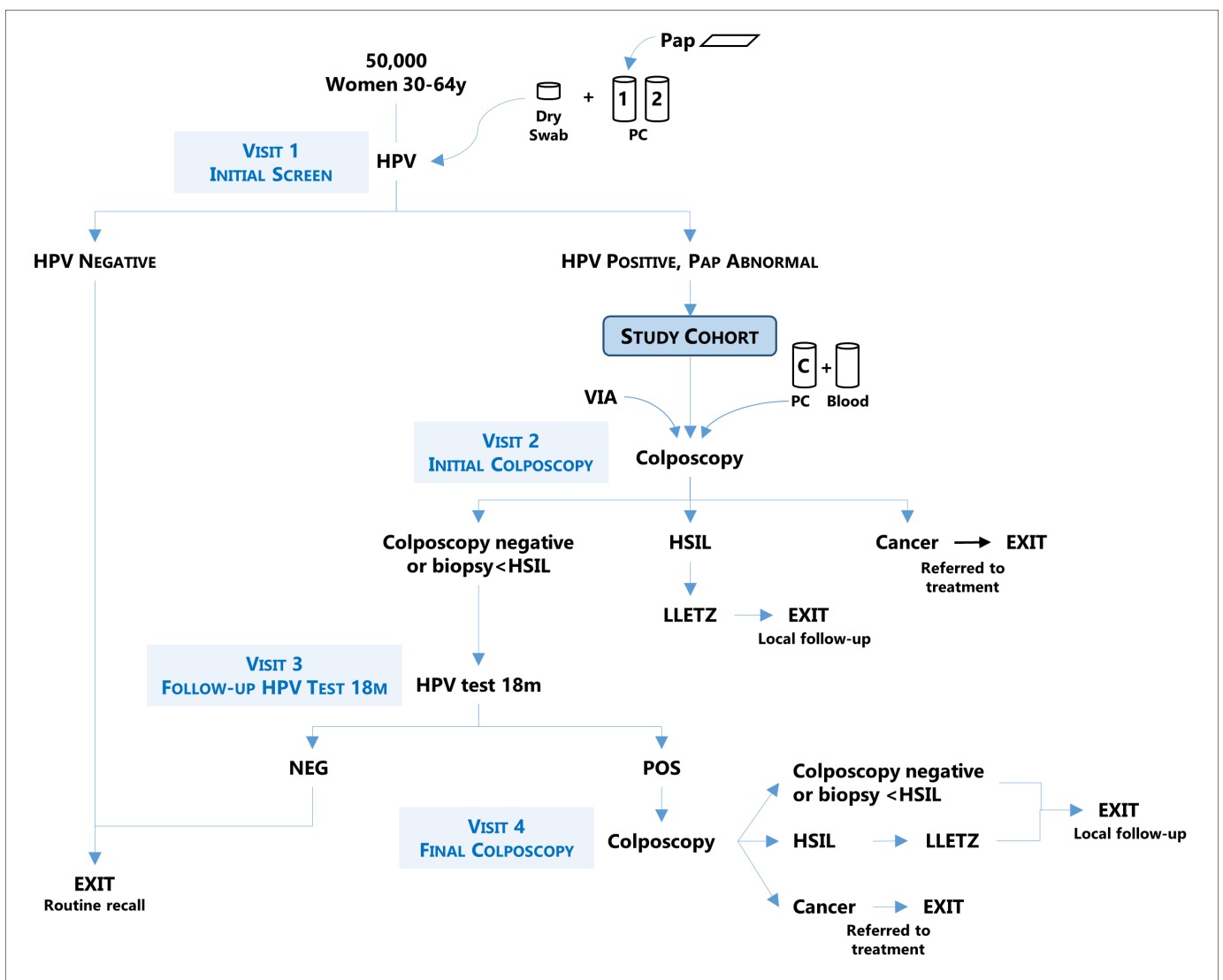

**Figure 1** Flowchart of study protocol. HPV, human papillomavirus; LLETZ, large loop excision of the transformation zone; VIA, visual inspection with acetic acid.

## Visit 1: initial screening

After obtaining informed consent and providing contact information and basic socio-demographic data to nurses or doctors trained on data confidentiality, women agreeing to participate are assigned a unique study ID number, and a clinician performs a pelvic exam and refers women to adequate care if clinically required. Next, after inserting a speculum without lubricant, the clinician collects cervical cells in the following order: (1) a Dacron swab that will be placed in a cryovial without preservation medium and frozen within 24 hours of collection; and (2) two consecutive collection brushes; the first brush is used to prepare a Pap smear and then washed in a vial with PreservCyt (Hologic) medium (PC); the second one is directly washed in a second PC vial. After being vigorously shaken inside the vials, the brushes are discarded and the vials are kept at room temperature or at 4°C until testing or aliquoting.

The frozen dry swab will be used for oncoprotein tests, the first vial for HPV testing and for LBC preparation, the remains and the second vial will be aliquoted for HPV testing quality control (QC) and other molecular tests under evaluation.

HPV testing is performed at local laboratories selected for the study. All HPV-positive women, those with abnormal cytology and those testing positive in any QC test become the study cohort, and are referred to colposcopy for diagnosis followed by treatment of cervical lesions if needed.

Women who are not referred to colposcopy are given their results, explanations on the significance of a negative HPV result and recommendations for future local regular care.

## Visit 2: colposcopy visit after abnormal screening results

At this visit, a risk factor interview, including sexual behaviour information, is administered.

A pelvic exam with collection of cervical cells using a swab that is washed in a PC vial is done before applying acetic acid to the cervix. This sample is used for a new HPV test and to perform other tests which could not be performed on the enrolment sample due to insufficient sample volume.

In some centres, after cervical sampling and immediately before colposcopy, a trained nurse, midwife or a doctor performs VIA. After application of 5% acetic acid, the clinician records the presence of acetowhite areas and documents his/her assessment on the need for treatment. The clinician does not discuss his/her findings with other team members to avoid inducing bias; except when cancer is suspected and the woman should be immediately referred to colposcopy.

After the VIA, the colposcopist inspects the cervix under the colposcope after application of acetic acid and collects 2–3 biopsies of all acetowhite areas observed. The colposcopist then ranks the biopsies according to the probability that the acetowhite area, from which each biopsy was collected, contained the worst lesion on colposcopic impression. No blind biopsies or biopsies from women without observable acetowhite areas are collected.

Collection of a 10 mL blood specimen from the arm is done at this visit, following usual procedures for blood collection. The blood sample will be used for HPV and cervical disease assays that could be available in the future, including markers of genetic susceptibility.

## Clinical management at first colposcopy

The clinical management of women attending colposcopy is defined by the enrolment cytology and the colposcopy results as shown in figure 2.

### Women with cytology HSIL or more severe lesion (HSIL+)

Treatment with LLETZ should be offered without prior histology confirmation to women with a 'positive major' colposcopy, whenever possible. Biopsies should be collected if the colposcopy is considered 'positive minor' or where LLETZ without confirmed CIN2 is not allowed. If a transformation zone type 3 (TZ3) is observed, an endocervical sample should be collected and an excision type 3 (LLETZ to excise lesions which endocervical extent is not visible) diagnostic or therapeutic should be performed.

The clinical management of women with discordant results: lesions less than CIN2 (<CIN2) or those with negative colposcopy but with HSIL+ cytology should be revised at multidisciplinary team (MDTs) meetings, that are attended by the cytologist, pathologist, colposcopist and the local principal investigator at least. During the MDT, cytology, histology and colposcopy results, considering the age and parity of the woman, should be discussed to finally recommend treatment or recall for a second HPV test 18 months later. In the benefit of women, who may be lost to follow-up, recommendation for LLETZ will be prioritised above 18 months recall.

### Women with cytology less than HSIL (<HSIL), unsatisfactory or unknown cytology

Biopsies should be collected from all women with positive colposcopy. An endocervical sample should be collected from women with TZ3 and if the result of this sample is HSIL+, treatment (excision type 2 or 3) should be offered. Women with histology <CIN2, negative colposcopy and those with a TZ3 and <HSIL on the endocervical sample should be recalled at 18 months. MDTs should be carried out when the colposcopy is positive major but the histology is <CIN2.

The colposcopic impression using a standard colposcopy nomenclature and specific colposcopy features (eg, size and location of observed lesions, number and severity of biopsies collected) are recorded in colposcopy study forms.

At the treatment visit, first a colposcopy is done followed by LLETZ. The reason and type of LLETZ as well as the colposcopic impression are recorded in treatment study forms. A pregnancy test is administered to participants before treatment with LLETZ and local consent forms

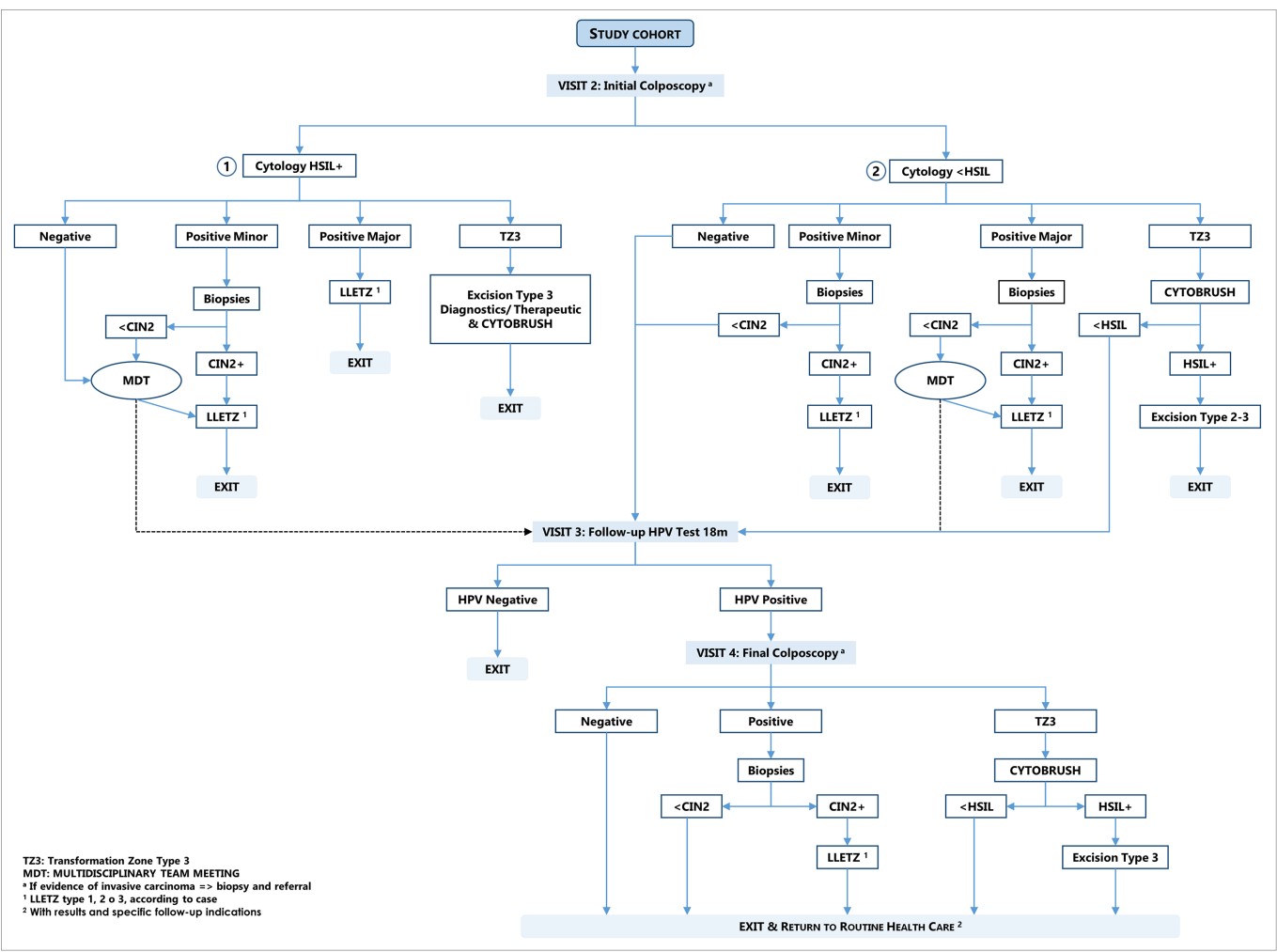

**Figure 2** Clinical management of women in the study cohort. CIN, cervical intraepithelial neoplasia; HPV, human papillomavirus; HSIL, high-grade squamous intraepithelial lesion; LLETZ, large loop excision of the transformation zone; MDT, multidisciplinary team.

for surgical procedures are sought. Self-reported adverse events are documented.

### Visit 3: follow-up HPV test at 18 months

All women who were not treated after an initial positive screen are invited to a final follow-up screening with HPV testing 18 months after enrolment. Women will be allowed to attend this visit up to 30 months after enrolment or until the study ends. To increase attendance to this visit, women may self-collect a vaginal sample either at the health centre or at home (depending on the follow-up strategy). If this was the case, women will be explained what self-sampling is and how to self-collect a sample using a graphical brochure. Women will be given either a *care*Brush (QIAGEN) to be inserted in a specimen transport medium to be tested with hybridisation techniques or a swab to be inserted into a tube with no preservation medium to be washed in PC and tested with PCR techniques. Alternatively, a clinician will perform a pelvic exam and collect the sample using a cytobrush to be washed in a PC vial. The clinician-collected sample will be used for HPV testing and future triage tests.

HPV-negative women exit the study and return to local regular screening. HPV-positive women will be referred for a final colposcopy round for diagnosis and treatment.

### Visit 4: final colposcopy

The clinical management of women attending final colposcopy will be defined only by the colposcopic impression. Biopsies will be collected if the colposcopy is positive and an endocervical sample will be collected if a TZ3 is observed. Women with local CIN2+ histology will be treated with LLETZ and those with HSIL+ endocervical cytology will be treated with excision type 3; afterwards these women exit the study and return to routine healthcare. Women with histology <CIN2 or endocervical cytology <HSIL or negative colposcopy also exit the study and return to routine healthcare.

All women exiting the study are given a report with their screening and diagnosis results and clear indications on how to continue with routine follow-up care. Study colposcopists who usually work on a hospital that covers the area selected for ESTAMPA (EStudio multicéntrico de TAMizaje y triaje de cáncer de cuello uterino

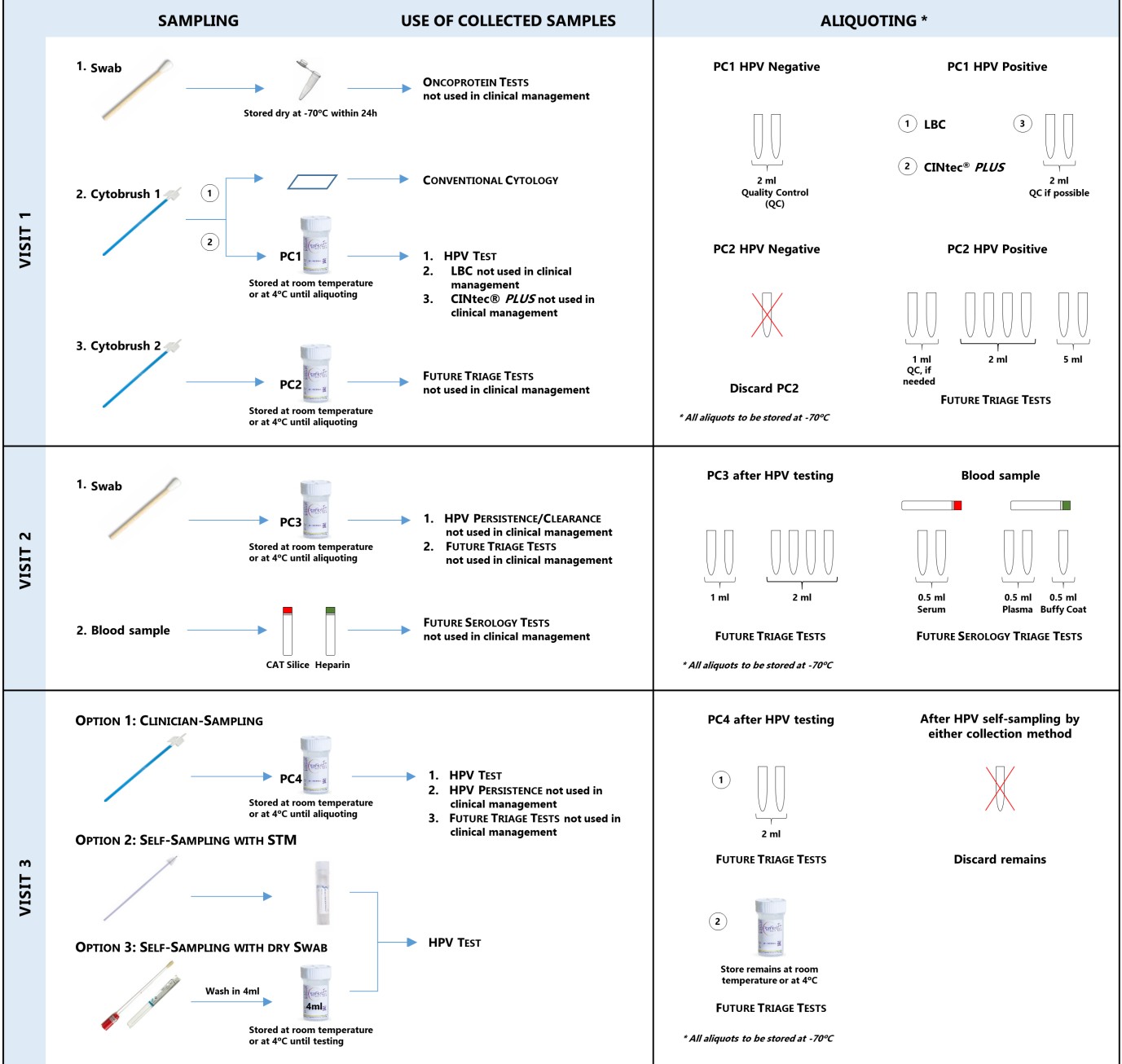

**Figure 3** Sample management and use. CIN, cervical intraepithelial neoplasia; HPV, human papillomavirus; LBC, liquid-based cytology; PC, PreservCyt; QC, quality control.

con pruebas del virus del PApiloma humano; Spanish acronym), have committed to follow-up treated women. It has been agreed that follow-up may be done by HPV testing or Pap with colposcopy of those HPV-positive or with abnormal smears, and that the first follow-up visit should be done at 6 or 12 months in accordance with local regular care.

### Sample management, use and biobanking

Figure 3 describes which samples and at which visit they are collected, which test they will be used for and which aliquots will be done and stored for future tests. Most triage tests will be evaluated using samples collected at

initial screening, to mimic a real-life situation of reflex testing. Additional samples will be collected at visits 2 and 3, mostly to evaluate HPV persistence and to use the remains for techniques that will become available in the future. The dry swab collected at initial screening is stored at −70°C until testing. PC vials are stored at controlled room temperature or 4°C before testing or aliquoting. Aliquots are stored at −70°C locally. Samples and aliquots are stored at each centre until they are used or transferred to a centre for specific centralised testing or to the International Agency for Research on Cancer (IARC) biobank for future laboratory work. Samples transferred to IARC

biobank are managed using the in-house laboratory information management system called Sample Management System for IARC (SAMI).

## Pathology

Cervical tissues collected by biopsy or LLETZ are fixed in buffered formalin at the colposcopy clinic and are transported at room temperature to a local pathology laboratory. Tissues are processed, cut and stained under standardised study procedures. Three cuts per tissue block are mounted into: (1) a regular H&E stained slide to be used for immediate histological diagnosis by a local pathologist and corresponding clinical management; (2) an electro-charged slide for p16 immunohistochemistry (IHC) stained at request and (3) a regular slide to be later H&E stained if needed.

The local study pathologist interprets the first H&E stained slide giving a diagnostic report under the CIN classification as follows: negative, atypical metaplasia, CIN1, CIN2, CIN3, AIS and invasive cancer (with morphological type: eg, squamous carcinoma, adenocarcinoma). All slides are stored at room temperature and will be sent to IARC for final study diagnosis by a review panel when required. Blocks are stored at each site but will be available for the study, in case if additional tests to confirm or clarify the study outcome are necessary.

## Study endpoints

Until recently, histologically confirmed cervical disease was classified as CIN grades 1, 2 and 3, with CIN3 being the most reproducible cancer precursor and CIN2 being an intermediate representing a mixture of HPV infections (CIN1) and cancer precursors (CIN3). A new nomenclature describing disease of the lower anogenital tract was proposed after a consensus exercise of a large number of international experts collaborating on the LAST project. The LAST classification incorporates the current knowledge of HPV biology and the use of biomarkers to improve diagnosis and recommends the use of $p16$ IHC to clearly define histological HSIL. Under LAST, cervical HSIL includes $p16$-positive CIN2, CIN3 and AIS.

The primary endpoint of the study is histologically confirmed HSIL or worse lesions (HSIL+). Two secondary endpoints will be considered: (1) locally diagnosed CIN2+ as CIN2 is the current treatment threshold and (2) locally diagnosed CIN3+ as CIN3 is more representative of cervical precancer under the CIN classification routinely used. Lesions detected after initial and follow-up screening will be counted as outcomes.

## Histology review and adjudication process

All study histology is reviewed by international experts on cervical pathology without knowledge of screening results. Reviewers may request to interpret $p16$ IHC slides in addition to the regular H&E, and will give results as follows: negative, LSIL, HSIL and cancer. As local histology is reported under the CIN classification: negative, CIN1, CIN2, CIN3 and cancer; concordance will be reached for the paired results: <CIN2 and <HSIL or CIN2+ and HSIL+ (local and reviewed, respectively).

The adjudication process involves two experts at a time. The first expert reviews the original diagnosis and only discordant diagnoses are reviewed by the second expert. Agreement diagnosis between the local and the first expert pathologist or between the two experts is considered adjudicated. When both experts do not agree, final adjudication is done in a face-to-face meeting, at which if agreement is not reached the second expert diagnosis prevails. For the purpose of the study, the final histology diagnosis could be <HSIL or HSIL+ (figure 4).

## Statistical analysis

Performance estimates (sensitivity, specificity, positive and negative predictive values) to detect histological HSIL+ for each triage candidate alone or in combination, and their potential to be used in primary screening whenever applicable, will be estimated (see table 1). McNemar's tests will be used to compare the proportion of women referred to colposcopy adjusting for multiple comparisons. Stratified analyses by age group (30–39, 40–49 and 50–64) and by study centre using the CIN2+ endpoint, will be done.

When evaluating tests in primary screening, relative measures of performance (eg, relative sensitivity of HPV testing vs cytology or vs cotesting) will be used.

Sensitivity analyses will be done to account for the impact of women not attending colposcopy or follow-up visits and those with inadequate screening results.

Challenges faced when implementing HPV-based screening and different approaches used to increase study participation and adherence to screening will be reported separately.

## Sample size calculation

The sample size is based on the ability to detect differences in sensitivities of a paired triage tests for the detection of the primary study endpoint (histology-based HSIL+) with a type I error of 0.05 according to Connor.[43] We conservatively assume that the population prevalence of HSIL+ is 1% based on previous studies.[44 45] We expect to screen about 50 000 women and to detect 500 HSIL+ cases. With this number of cases, the study will have 80% power to detect a 5% difference in sensitivities between two triage tests for pairwise discordances up to 10%, or 90% power for pairwise discordance up to 8%. Details are presented in supplementary material (online supplementary file 1: sample size calculation).

## Data capture system

Clinical and laboratory data are collected at study centres using standardised paper forms that are entered into a centralised web-based system specifically developed for the study in Spanish, with appropriate security, privacy and automated backup system. The capture is standardised but allows site-specific customisation. The information system also monitors timely implementation, quality of

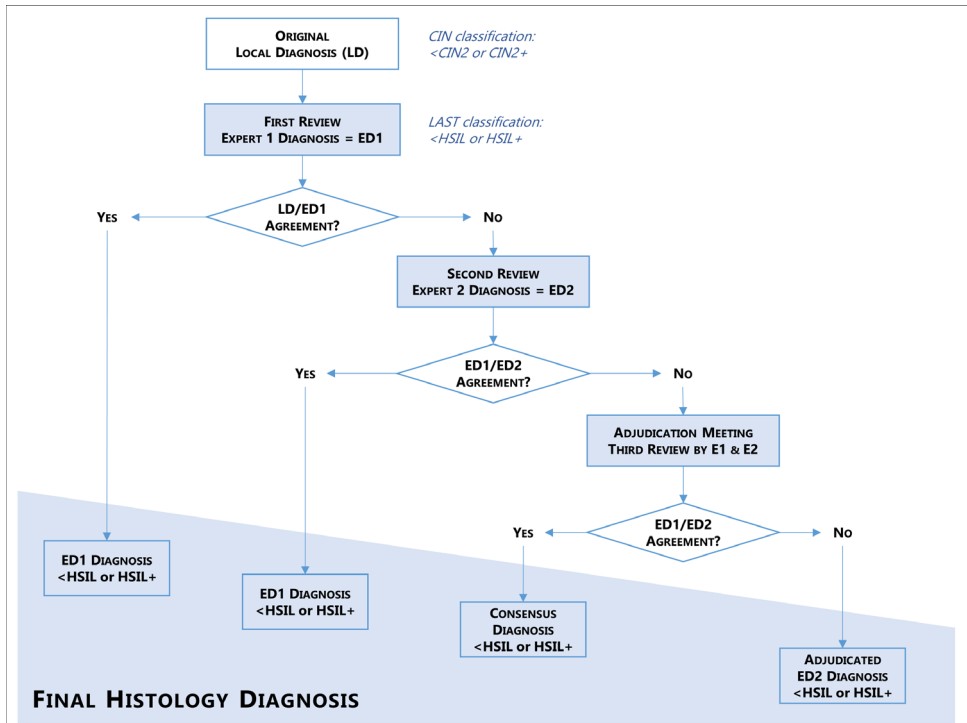

**Figure 4** Study endpoint adjudication process. CIN, cervical intraepithelial neoplasia; HSIL, high-grade squamous intraepithelial lesion.

| Table 1 | Definition of accuracy measures of screening tests | |
|---|---|---|
| | **Numerator** | **Denominator** |
| **Tests for triage of HPV-positive women** | | |
| Sensitivity | No. HPV-positive women testing positive on triage with disease* | Primary: No. HPV-positive women with HSIL+ Secondary: No. HPV-positive women with CIN2+; No. HPV-positive women with CIN3+ |
| Specificity | No. HPV-positive women testing negative on triage with no disease† | Primary: No. HPV-positive women with <HSIL Secondary: No. HPV-positive women with <CIN2 |
| PPV | No. HPV-positive women testing positive on triage with disease* | No. HPV-positive testing positive on triage |
| NPV | No. HPV-positive women testing negative on triage with no disease† | No. HPV-positive testing negative on triage |
| **Tests for primary screening** | | |
| Relative sensitivity (HPV/cytology) | No. HPV-positive with disease* | No. abnormal cytology with disease* |
| Relative specificity (HPV/cytology) | No. HPV-negative with no disease† | No. cytology NILM with no disease† |
| Relative sensitivity (HPV/cotesting) | No. HPV-positive with disease* | No. HPV-positive OR abnormal cytology with disease* |
| Relative specificity (HPV/cotesting) | No. HPV-negative with no disease† | No. HPV-negative AND cytology NILM with no disease† |

*Women with disease: women with HSIL+ on review (primary endpoint) or women with local CIN2+ or CIN3+ (secondary endpoints).
†Women with no disease: women with negative, CIN1, LSIL histologic diagnosis, HPV-negative women at 18 months and women with final (18 months) negative colposcopy.
CIN2, CIN grade 2; <CIN2, histological diagnosis less than CIN2: negative, CIN1; CIN2+, CIN2 or worse lesions; CIN3, CIN3 grade 3; CIN3+, CIN3 or worse lesions; CIN, cervical intraepithelial neoplasia; HPV, human papillomavirus; <HSIL, histological diagnosis less than HSIL: negative, HSIL; HSIL, high-grade squamous intraepithelial lesion on histology; HSIL+, HSIL or worse lesions; NILM, Negative for intraepithelial lesions or malignancy; NPV, negative predictive value; PPV, positive predictive value.

 Almonte M, *et al. BMJ Open* 2020;**10**:e035796. doi:10.1136/bmjopen-2019-035796

inputs, progress and highlight activities where special attention is needed to guarantee the study outcome. All data are treated as confidential and kept for as long as required by law.

## Patient and public involvement

There was no patient or public involvement in the design of this trial.

## Ethical considerations

The study protocol was approved by the Ethics Committee of the International Agency for Research on Cancer (IEC Project 12–27-A7), the Pan American Health Organization (PAHO) Ethical Committee and Ethical Committees in each of the study participating centres (online supplementary file 2: list of ethical committees that have approved the study). The current version of the protocol was approved by IEC this year (V.3.2, revised on 17 January 2018). The informed consent includes details on the background, procedures of the study, risks and benefits, statement of confidentiality, specimen use and study staff to contact.

The study is considered minimal risk as the procedures are standard practice in cervical cancer screening programmes.

A Data and Safety Monitoring Board (DSMB) has been established to monitor progress of the study, assure participant safety, advice on scientific conduct and analysis of the study and suggest improvements or modifications to the protocol. The DSMB is formed by international experts on HPV infection, cervical cancer and screening: a gynaecology oncologist, a medical public health specialist, an epidemiologist and a statistician, as well as, two Latin American women: a medical scientist and women's rights advocate.

## Dissemination

Scientific reports on each triage candidate and of triage combinations, using all study data, will be published in peer-reviewed scientific journals. The DSMB will advise on the evaluation of novel emerging technologies as the study goes on. In addition, support will be given to local investigators to propose and lead analyses that could use all study data, data from one centre or from several centres.

## DISCUSSION

The ESTAMPA study represents a large collaboration organised among Latin American cervical cancer researchers to jointly contribute to cervical cancer prevention. About 50 000 women in the region will be screened, many of those for the first time, with a highly sensitive HPV test with efforts concentrated on treating all women with detected HSIL+.

The study will contribute to establish the clinical management of HPV-positive women under different scenarios, those where high-tech molecular biomarkers can be used as triage and those where low-tech VIA may be the only suitable triage test.

The introduction of HPV testing in primary screening is imminent in the region. In fact, Argentina and Mexico have been offering HPV screening for several years within the public health system. In the other countries, HPV testing is available in the private sector and is slowly becoming available or there are plans to introduce it at

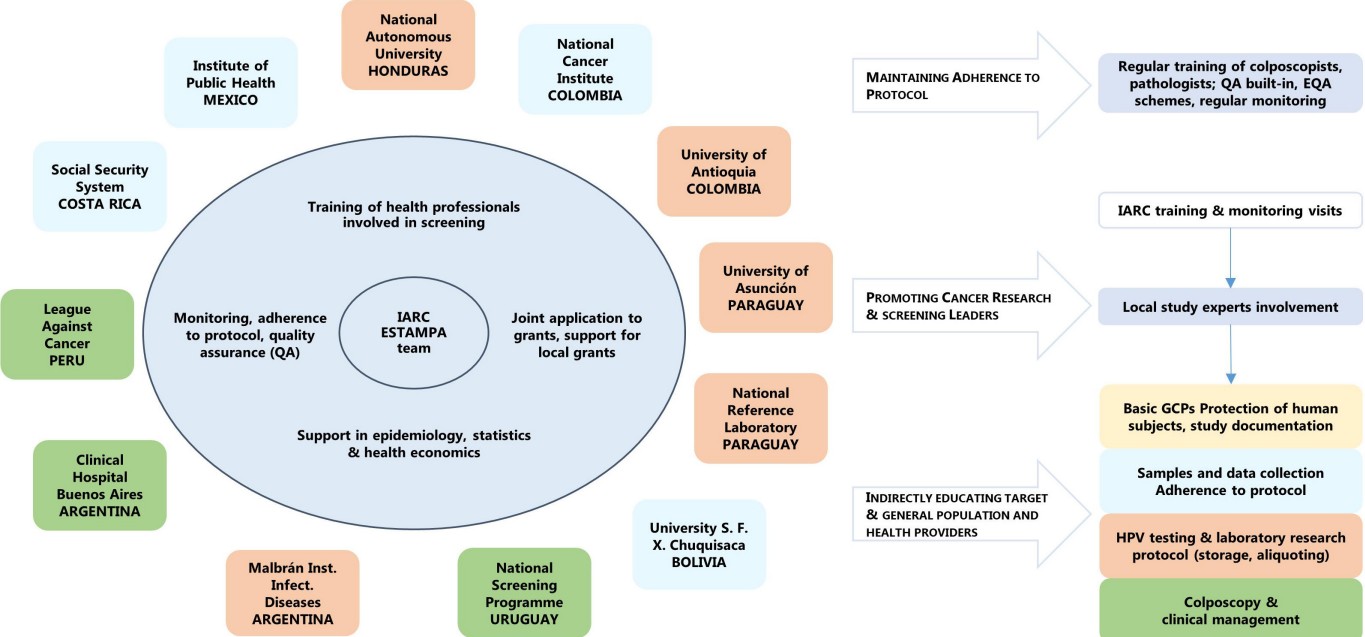

**Figure 5** ESTAMPA study network. EQA, External Quality Assessment; GCPs, Good Clinical Practices; IARC, International Agency for Research on Cancer.

reduced cost in public systems. Clinical guidelines on how to follow-up HPV-positive women are under development, and the use of cytology as triage as done in HIC need to be proven acceptable in Latin America, where cytology has demonstrated to have very low sensitivity and attempts to organise cytology-based screening programmes have largely failed. More sensitive and affordable triage tests to be used in Latin America are awaited, the study will contribute with evaluating the most promising ones.

A few limitations of the study design should be mentioned. First, capacity (personnel and facilities) differences among study centres could influence results. To prevent this, special attention is given to training health professionals involved in the study: clinicians collecting samples, colposcopists, pathologists, molecular biologists among others. The training includes good clinical practices, research governance and up-to-date specialty training to standardise procedures all over the study. Second, we are not performing colposcopy nor collecting biopsies in HPV-negative women, potentially introducing verification bias for evaluation of techniques in primary screening but not in triage. However, as cotesting with Pap is being done, and although the rate of cytological abnormalities among HPV-negative women will be usually low, we will have a small group of them having colposcopy helping us to estimate disease among HPV negatives. We also do not collect random biopsies at colposcopy when acetowhite lesions are not evident, thus, it is possible that some lesions are being missed. To minimise missing lesions, the study protocol includes: (1) 2–3 biopsies of any acetowhite areas observed are collected at colposcopy with ranking of severity of biopsies collected to further evaluate the use of multiple and severity of biopsy collection; and (2) a second round screening at 18 months for women with no evident disease at initial screening, providing an additional opportunity of detecting any missed disease at initial screening.

The main strengths of our study are: (1) the design will allow the evaluation of a series of triage techniques without influencing the outcome of the study, as tests are performed/evaluated usually after cervical disease status of HPV-positive have been determined; (2) the sample size will allow the evaluation of multiple combinations of techniques and to adjust for centres heterogeneity when needed; (3) the study is centrally coordinated by IARC with participation of 12 study centres, each of them with strong expertise in different cervical screening aspects (figure 5) and opportunity has been given to junior investigators to participate in training and monitoring missions, thus, contributing to promote new cancer research Latin American leaders and to consolidate a large network of screening professionals; and (4) the multicentric nature of the study will also allow capturing experiences from areas which are geographically, culturally and socioeconomically distinct from each other and with different health systems/areas that may face common challenges but that require different approaches in accordance with their context.

It is also important to highlight that the study is mostly being run within public health services already in place, with the exception of HPV testing that has been implemented in some university or hospital laboratories for the study. Thus, the study will substantially contribute to further scale-up of HPV testing as recommended by Pan American Health Organization by developing a 'screening platform' for implementation of HPV-based cervical screening programmes in the future.

**Author affiliations**
[1]Prevention and Implementation Group, International Agency for Research on Cancer, Lyon, Rhône-Alpes, France
[2]Pontificia Universidad Javeriana, Bogotá, Colombia
[3]Grupo de Infección y Cáncer, Universidad de Antioquia, Medellín, Colombia
[4]Agencia Costarricense de Investigaciones Biomédicas (ACIB), Fundación Inciensa, Guanacaste, Costa Rica
[5]Instituto de Investigaciones en Microbiología, Universidad Nacional Autónoma de Honduras (UNAH), Tegucigalpa, Honduras
[6]Instituto Nacional de Enfermedades Infecciosas - ANLIS Malbrán, Buenos Aires, Argentina
[7]Instituto Nacional de Cancerología, Bogotá, Colombia
[8]Instituto de Salud Pública de México, Morelos, México
[9]Liga contra el Cáncer-Perú, Lima, Perú
[10]Advanced Center for Chronic Diseases, ACCDiS, Facultad de Medicina, Pontificia Universidad Católica de Chile, Santiago, Chile
[11]Instituto de Investigaciones en Ciencias de la Salud, Universidad Nacional de Asunción, San Lorenzo, Paraguay
[12]Comisión Honoraria de Lucha contra el Cáncer, Montevideo, Uruguay
[13]Caja Costarricense de Seguro Social (CCSS), Región Pacífico Central, San José, Costa Rica
[14]Clínica Angloamericana, Lima, Perú
[15]Escuela de Medicina Humana, Universidad de Piura, Lima, Perú
[16]Laboratorio Central de Salud Pública, Asunción, Paraguay
[17]Hospital de Clínicas José de San Martín, Buenos Aires, Argentina
[18]Facultad de Medicina, Universidad Mayor, Real y Pontificia de San Francisco Xavier de Chuquisaca, Sucre, Bolivia
[19]SMS-Oncology, Amsterdam, The Netherlands
[20]Screening Group, International Agency for Research on Cancer, Lyon, Rhône-Alpes, France
[21]Centro de Estudios de Estado y Sociedad/Consejo Nacional de Investigaciones Científicas y Técnicas, Buenos Aires, Argentina
[22]Hospital Dr Enrique Baltodano Briceño, CCSS, Liberia, Costa Rica
[23]Hospital Nacional, Ministerio de Salud Pública y Bienestar Social, Itauguá, Paraguay
[24]Instituto Nacional del Cáncer, Ministerio de Salud Pública y Bienestar Social, Capiatá, Paraguay
[25]Centro de Salud Ciudad de la Costa, ASSE, Ciudad de la Costa, Uruguay
[26]Hospital Policial, DNASS, Montevideo, Uruguay
[27]Hospital de Clínicas, Facultad de Medicina, UDELAR, Montevideo, Uruguay
[28]Hospital Nacional Profesor Alejandro Posadas, Buenos Aires, Argentina
[29]Patología Oncológica SAC, Lima, Perú
[30]Instituto Colombiano de Medicina Tropical Antonio Roldán Betancur, Universidad CES, Apartadó, Colombia
[31]Secretaría de Salud, Tegucigalpa, Honduras
[32]Laboratorio de Biología Molecular, Departamento de Patología Clínica, Centro Hospitalario Pereira Rossell, Montevideo, Uruguay
[33]Hospital Dr Rafael Angel Calderón Guardia, CCSS, San José, Costa Rica
[34]Instituto de Patología y Biología Molecular, Lima, Perú
[35]ESE Hospital Antonio Roldán Betancur, Apartadó, Colombia
[36]Hospital Materno Infantil de San Lorenzo, Ministerio de Salud Pública y Bienestar Social, San Lorenzo, Paraguay
[37]Programa Nacional contra el Cáncer, Tegucigalpa, Honduras
[38]Departamento de Anatomía Patológica y Citología, Hospital de la Mujer, Centro Hospitalario Pereira Rossell, Montevideo, Uruguay
[39]Hospital Monseñor Víctor Manuel Sanabria Martínez, CCSS, Puntarenas, Costa Rica

[40]Hospital Materno Infantil de Trinidad, Ministerio de Salud Pública y Bienestar Social, Asunción, Paraguay
[41]Hospital Gineco-Obstétrico y Neonatal "Dr Jaime Sánchez Porcel", Sucre, Bolivia
[42]Pan American Health Organization (PAHO), Washington, District of Columbia, USA
[43]Department of Sexual and Reproductive Health and Research, World Health Organization, Geneva, Switzerland
[44]Department of Pathology, University of California, San Francisco, California, USA

**Contributors** MA, RM, GIS, PG, AF, MAP, CW, ACV, ELP, JJ, CF and RH conceptualised the study. MA and RH are the principal investigators responsible for overall conduct of the study. GIS, AF, MAP, CW, ACV, EK, LM, GR, AC, GV, VV, ST, LF and CT are the local principal investigators responsible for recruitment, clinical management and data collection. MA and AB provide statistical support and together with MLH, MLR, EL, SB and ATR centrally coordinate the study. MA, PG, CW, SA and MIR conceptualised the psychosocial and implementation aspects of the study. MA, AB, MLR, EG and TD conceptualised the histology review process. MA, PG, MAP, JJ, LM, GV, ST, LF, AB, MLH, MLR, EL, EG, MC, SM, YS, MO, AVB, NP, MRP, MR, PHN, SL, NB, TD and RH contributed to local capacity building. SM, YS, MAN, GIS, LM, MIR, YC, AF, BS, LG, AC, MAR, MCC, MAP, JAS, YBF, MR, GV, MLB, GO, IBG, PHN, AMH, LF, LC, BF and JP coordinate recruitment and laboratory procedures. MC, JMB, EK, MO, AS, JF, AVB, BC, NP, EG, CS, AM, JM, MRP, FD, GV, DG, HR, LC, LF, ST and OL coordinate colposcopy and pathology procedures. MA wrote the article with the support of RH. RM, GIS, JJ, CF, AB, MLR, ATR and TD contributed to finalisation of the manuscript. All authors reviewed and approved the final version of the manuscript.

**Funding** This project acknowledges the financial contribution from IARC, the UNDP (United Nations Development Program)–UNFPA (United Nations Population Fund)–UNICEF–WHO–World Bank Special Program of Research, Development and Research Training in Human Reproduction (HRP), a co-sponsored program executed by the Department of Sexual and Reproductive Health and Research of the World Health Organization (HRP/WHO), the Pan American Health Organization, the National Cancer Institute (NCI) Center for Global Health, the National Council for Science and Technology (CONACYT) from Paraguay; National Cancer Institute of Argentina, National Cancer Institute of Colombia, Costa Rica Social Security Fund (CCSS), and all local collaborative institutions.

**Disclaimer** Where authors are identified as personnel of the International Agency for Research on Cancer/WHO, the authors alone are responsible for the views expressed in this article and they do not necessarily represent the decisions, policy or views of the International Agency for Research on Cancer / WHO.

**Competing interests** None declared.

**Patient and public involvement** Patients and/or the public were not involved in the design, or conduct, or reporting or dissemination plans of this research.

**Patient consent for publication** Not required.

**Provenance and peer review** Not commissioned; externally peer reviewed.

**ORCID iD**
Maribel Almonte http://orcid.org/0000-0003-1623-8323

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
