## [Reviewer comments · BMJ Open]

ARTICLE DETAILS

TITLE (PROVISIONAL)	Multicentric study of cervical cancer screening with human papillomavirus testing and assessment of triage methods in Latin America: the ESTAMPA screening study protocol
AUTHORS	Almonte, Maribel; Murillo, Raul; Sanchez, Gloria; González, Paula; Ferrera, Annabelle; Picconi, Maria Alejandra; Wiesner, Carolina; Cruz-Valdez, Aurelio; Lazcano-Ponce, Eduardo; Jerónimo, Jose; Ferreccio, Catterina; Kasamatsu, Elena; Mendoza, Laura; Rodríguez, Guillermo; Calderón, Alejandro; Venegas, Gino; Villagra, Veronica; Tatti, Silvio; Fleider, Laura; Teran, Carolina; Baena, Armando; Hernandez, María de la Luz; Rol, Mary Luz; Lucas, Eric; Barbier, Sylvaine; Ramirez, Arianis Tatiana; Arrossi, Silvina; Rodríguez, Maria Isabel; González, Emmanuel; Celis, Marcela; Martínez-Better, Sandra; Salgado, Yuly; Ortega, Marina; Beracochea, Andrea Veronica; Pérez, Natalia; Rodríguez de la Peña, Margarita; Ramón, María; Hernández, Pilar; Arboleda, Margarita; Cabrera, Yessy; Salgado, Brenda; García, Laura; Retana Peña, Marco; Colucci, Maria Celeste; Javier Arias, Stella; Bellido-Fuentes, Yenny; Bobadilla, Maria Liz; Olmedo, Gladys; Brito, Ivone; Mendéz-Herrera, Armando; Cardinal, Lucía; Flores, Betsy; Peñaranda, Jhacquelin; Martínez, Josefina; Soilán, Ana; Figueroa, Jacqueline; Caserta, Benedicta; Sosa, Carlos; Moreno, Adrián; Mural, Juan; Doimi, Franco; Giménez, Diana; Rodríguez, Hernando; Oscar, Lora; Luciani, Silvana; Broutet, Nathalie; Darragh, Teresa; Herrero, Rolando

VERSION 1 – REVIEW

REVIEWER	Alejandra Castanon Kings College London, United Kingdom
REVIEW RETURNED	16-Dec-2019

GENERAL COMMENTS	Many thanks for the opportunity to read the ESTAMPA study protocol. This is a very ambitious project which should provide critical data allowing Latin-American countries evidence on which to proceed with the introduction of HPV based cervical screening programmes. The project is forward thinking in its scope to evaluate many triage options and new technologies going forward. The training and EQA scheme appear to be a key component in ensuring that the study leaves a legacy for individual programmes to build on. Please find my comments below. Major comments 1.- Please clarify that the primary screening test will be a co-test. The word co-test is only used in the strengths and limitations summary after the abstract. The manuscript would benefit from the term being included in the first sentence of study design (i.e. This is a screening study in which HPV testing and Pap (co-testing) will be ...). Also in the abstract, first sentence methods and analysis. I've
--

	also made some comments on Figure 1 relating to this (attached as pdf). 2.- A secondary objective of the study is assessment of the feasibility of implementing organised HPV-based screening programmes in LA. I found that the creation of evidence to support this point was not very detailed. The data capture system seems key in achieving this aim and in this reviewer's opinion the paragraph devoted to it did not make clear the role it has to play in scaling up screening. For example is this system the "screening platform" mentioned in the discussion (last paragraph page 22)? 3.- Further how are you evaluating colposcopy capacity and whether the training programme has improved the quality of your colposcopy – a key ingredient for organised screening? 4.- Figure 3 – visit 3. Why are the remains of the self-sample discarded and not stored for future potential biomarkers (an area which is quickly developing)? It seems strange that the clinician samples are stored but not the self-samples. Minor comment - Last bullet point of strengths and limitations of this study. Would it be clearer if it said something such as "colposcopy and collection of biopsies was not performed in all negative women (only in a subset who were cytology positive), potentially"
--	---

REVIEWER	Kimon Chatzistamatiou Aristotle University of Thessaloniki, Thessaloniki, Greece
REVIEW RETURNED	27-Jan-2020

GENERAL COMMENTS	The submitted article describes the study protocol of the ESTAMPA study, a multicentric study of cervical cancer screening with human papillomavirus testing and assessment of triage methods, conducted in Latin America. The subject is very important since the advent of primary HPV screening, worldwide, makes the identification of accurate triage methods for the HPV positive women imperative. The study is well designed so that it can address the research hypotheses set by the investigators, and the sample size is adequate. It is also important that during the study the investigators will form a biobank useful for examining novel triage methods in the future. The methods are described in detail and the limitations of the study are addressed.
---

REVIEWER	MA Sciensano, Brussels
REVIEW RETURNED	01-Feb-2020

GENERAL COMMENTS	GENERAL COMMENTS Given the high level of evidence that secondary prevention of cervical cancer using HPV-tests protects better than cytology, more and countries are switching to HPV-based screening. However, since the cross-sectional specificity of HPV testing is lower than cytology, appropriate triage of HPV-positive women is crucial. Almonte et al engaged in an ambitious plan to investigate the accuracy of several triage markers nested in a large screening study set up in Latin America. The ESTAMPA has the potential to become a landmark study which may trigger introduction of HPV screening in this continent and contribute to the WHO goal to make cervical cancer a rare disease; The study recruits 50,000 women for screening among whom ~5000
---

women might be hrHPV+. If only ~5000 hrHPV+ women are used to address triage questions, the investment of enrolling such a large screening population is considerable. Finding ~5000 hrHPV+ women in an existing screening programme would look more efficient. It might be useful to add a hypothesis regarding the accuracy of the used screening tests (for instance CIN2/3+ detection rate ratios cyto vs hpv vs contesting.. Another justification could be: given absence of HPV-based screening in Latin America, enrollment of a large screening population is needed in order to find sufficient hrHPV+ women where triage hypotheses could be assessed with sufficient power.

The pages are not numbered, the line numbering is not adjusted to the line spacing and only the last digit of the line numbering is visible. This complicates the review process.

It would be much clearer for the reader if the CIN nomenclature should be applied as only terminology to describe severity of cervical lesions in biopsies. SIL could be reserved to describe cervical cytology. LAST is very confusing and always needs adjectives "cytological" or "histological" and even requires definition of the topography "cervical". Using the CIN and SIL nomenclature throughout the whole paper would increase the clarity and reduce the word counts.

I assume that the authors can easily address the given suggestions in a subsequent manuscript.

SPECIFIC COMMENTS

TITLE

No comments.

ABSTRACT

Introduction could be shorter

In spite of screening 50,000 women no objective regarding screening is mentioned. Only the triage objectives are mentioned.

Methods and analysis

"with biopsy and treatment" "undergo colposcopy with biopsy and treatment". Explain the conditions for biopsy and treatment".

The sample size could be mentioned (50 000 women screened, expected 500 CIN2+ cases).

STRENGTHS AND LIMITATIONS

4th bullet. Unclear since the ESTAMPA does not include an intervention to increase screening coverage.

5th bullet. If the purpose of ESTAMPA would be to assess the absolute sensitivity and specificity, verification bias would be an issue. However for triage of hrHPV positive women, verification bias is not a problem. Even the assessment of the relative sensitivity of screening with cytology vs HPV vs cotesing would not really be affected by verification bias.

INTRODUCTION

Line 2, ref 1. The following ref (PM:31812369) is more specifically describing the current burden of CC.

Line 12. Ref for effective and safe vaccines (PM:29740819).

	Near end of 2nd § of introduction. Ref for increase uptake by offering self-sampling (PM:30518635). MATERIALS AND METHODS The study questions could be defined more concretely. A table could be added which defines the study's intermediate and final endpoints, with numerators and denominators. Which HPV test will be used for screening? What will be the conservation medium/media used for the screening/triage tests? PreservCyt only mentioned in the BIOBANKING §. This could be defined earlier. The computation of the sample size could be better documented. It would be useful to add, in Suppl materials, all elements included in the formula as well as the formula (with reference for the formula). Page with title "VISUAL METHODS on line 3" Sentence on top: add "hrHPV and type-specific" before "persistence". Third last sentence HPV persistence as a triage strategy Only 1-3 months is very short to assess persistence. Can this not be defined at 3-6M? Last §. Line 54. "Some of the methods already...". Replace by: "Some of the screening HPV tests already ...". Replace "individual" by "separate". METHYLATION Note the enormous heterogeneity in target genes (viral or human) and detection systems currently described. PARTICIPANTS Exclusion of of patients with treated precancer (6 months). Period is quite short. The first post treatment examination often is at 6M. Consider a longer exclusion period: fi 9 M. STUDY DESIGN Line 3 referred to colposcopy with biopsy. Add "if needed (see below)" The treatment visit is not mentioned. Also post-treatment follow-visit(s) is/are not mentioned VISIT 1 Line 32. Two consecutive brushes (which brushes?). Note that the endocervical brush (fi Cytobrush) is inappropriate for cytology (Cervical broom or combination with extended spatula) is recommended. It would be better to use only one term for cytology "LBC" instead of Pap smear which usually means conventional cytology. Line 45 "those testing positive in any QC test become the study cohort". If the women testing at QC (colpo visit), then the persistence cannot be assessed anymore. Consider not including the QC results in the study cohort or to include the women concerned only in a sensitivity analysis.
--	--

	Page with CLINICAL MANAGEMENT AT FIRST COLPOSCOPY Line 11. "to avoid inducing bias". Replace by "to avoid breaching the principle of independence of the outcome assessment" After the VIA, the colposcopist inspects. I assume it will be another colposcopist than VIA inspector. In this case "another" should be added before colposcopist. VISIT 3; Self-collection. Hybridisation techniques such as cHPV and HC2 should be avoided since proven to be significantly less sensitive and specific for CIN2+ and CIN3+ on self-samples compared to on clinician-taken samples. Only validated PCR methods should be used on self-samples (Ref PM:30518635) Line 41. Cytobrush is insufficient for a representative cytology (see above). PATHOLOGY Last §. CIN will be used for histology classification. This is an excellent terminology. As remarked earlier, for reasons of consistency and clarity CIN should be used as only terminology throughout the paper. Page with SAMPLE SIZE CALCULATION See remark in General Comments (above). For the assessment of the accuracy of triage of hrHPV+ strategies/markers, verification bias is not an issue. Cases with CIN2+ found only by cytology are not part of the Triage Cohort. (see above). Last sentence in first §. Challenges regarding participation. This sentence is very vague. Is there an indicator to assess participation? Follow adherence, treatment compliance could be easily assessed from the study design. DISCUSSION Last page. Line 4 "could mask final results". Better; "could influence results". How was multiple testing and inter-centre heterogeneity included in the sample size computation. Can be detailed in Suppl methods (see above).
--	--

VERSION 1 – AUTHOR RESPONSE

Reviewer: 1

Reviewer Name: Alejandra Castanon

Institution and Country: Kings College London, United Kingdom

Please state any competing interests or state 'None declared': None Declared

Please leave your comments for the authors below

Dear Editor,

Many thanks for the opportunity to read the ESTAMPA study protocol. This is a very ambitious project which should provide critical data allowing Latin-American countries evidence on which to proceed with the introduction of HPV based cervical screening programmes. The project is forward thinking in its scope to evaluate many triage options and new technologies going forward. The training and EQA scheme appear to be a key component in ensuring that the study leaves a legacy for individual programmes to build on. Please find my comments below.

Major comments

1.- Please clarify that the primary screening test will be a co-test. The word co-test is only used in the strengths and limitations summary after the abstract. The manuscript would benefit from the term being included in the first sentence of study design (i.e. This is a screening study in which HPV testing and Pap (co-testing) will be). Also in the abstract, first sentence methods and analysis. I've also made some comments on Figure 1 relating to this (attached as pdf).

Response:

The study has been designed to use HPV testing in primary screening and not co-testing. Conventional cytology was added because in most Latin American countries HPV testing is not included in screening guidelines or if included is not yet widely available. We implemented the preparation of cytology from the first cervical sample collected, thus, the Cervex brush is smear into glass slide ahead of being washed in PreservCyt medium, so that cytology could be assessed as if used in reflex testing.

However, we understand the concern and furthermore the third reviewer, Dr Arbyn, has suggested to estimate relative measures of performance such as: HPV testing versus cytology or versus co-testing to evaluate the performance of tests in primary screening, to avoid obtaining biased estimates due to no referral to colposcopy of negative screenees.

Nevertheless, we would like to emphasise that the study was not designed to evaluate co-testing, but having done Pap smears in women we will be able to evaluate: 1) cytology as triage of HPV positive women, as in reflex testing, 2) triage tests in women with negative cytology, and, 3) relative measures of performance. This has also been added in a sentence on the Statistical Analysis Section (Page 17, lines 4-5).

2.- A secondary objective of the study is assessment of the feasibility of implementing organised HPV-based screening programmes in LA. I found that the creation of evidence to support this point was not very detailed. The data capture system seems key in achieving this aim and in this reviewer's opinion the paragraph devoted to it did not make clear the role it has to play in scaling up screening. For example is this system the "screening platform" mentioned in the discussion (last paragraph page 22)?

Response:

We agree that this has not been fully explained in this manuscript. We decided to concentrate in this manuscript on the first main objective of triage evaluation due to the word limit. Thus, the methodology used and results related to the implementation objective will be reported separately.

However, several efforts have been carried out to ensure the development of a screening platform that can be later replicated and support the scaling-up of screening.

First, our study have followed essential steps to start a screening programme:

- i. selecting a target population (women 30-64y, area to covered),
- ii. using public health services as much as possible
- iii. building local capacity through training of health providers in screening activities (laboratory, primary care, secondary care)
- iv. generating regional networks of health professionals that can lead future implementation,

- v. applying different approaches to invite and inform women about cervical screening and to increase screening uptake,
- vi. selecting a test (according to local availability and capacity),
- vii. recording information needed to evaluate the process using of a data-capture system for registration and monitoring of screening activities
- viii. establishing adequate turnaround of screening results,
- ix. establishing optimal logistics to transport, store and manage samples,
- x. devising approaches to ensure adherence to colposcopy, treatment and other follow-up visits, and,
- xi. engaging stakeholders on understanding the overall screening process, on how to improve it and scale-up in the future.

Second, in preparedness for roll out of HPV testing, we developed a scale to assess the negative impact of receiving a positive HPV result among Latin American women. This tool was validated in Colombia and Argentina (*Psycho-social impact of positive Human Papillomavirus testing in Jujuy, Argentina, results from the Psycho-ESTAMPA study; Arrossi S, Almonte M, Herrero R et al, Preventive Medicine Reports, in press*), and will be used in a second-phase of ESTAMPA at which women will be recalled to attend a five-year screening visit as expected to happen in organised HPV-based screening. This second phase will be dedicated to study the implementation process using mixed methods approaches in countries ready for large-scale HPV-based cervical screening such as Colombia, Uruguay and Costa Rica.

Finally, we have conducted a survey to explore the acceptability and feasibility of implementing HPV testing across health providers involved in the screening process (those working on primary care clinics, colposcopy clinics and laboratories). Results from this survey will also be reported separately.

We have added the following sentence in the Methods Section (Page 7, lines 14-15), immediately after the objective:

“The methodology of used for this objective will be reported separately”.

3.-

Further how are you evaluating colposcopy capacity and whether the training programme has improved the quality of your colposcopy – a key ingredient for organised screening?

Response:

We did not design the study to evaluate the performance of colposcopy. However, we have been taken measures to standardise colposcopy across study centres. Briefly:

- 1) The ESTAMPA study has mainly been conducted in medium to large cities in Latin America. In these cities, there are usually a large number of colposcopists, colposcopy societies or gynaecology societies with colposcopy subgroups. All ESTAMPA centres were asked to identify experienced local colposcopists ahead of starting recruitment.
- 2) As the study was rolled out centre by centre, the first three colposcopists spent a week at the National Cancer Institute of Colombia for initial training on the clinical management protocol.
- 3) Further on, annual meetings of colposcopists already participating in the study as well as those collaborating with potential study centres and others from the Latin American region proposed by the UICC, PAHO, PATH and other regional organisations.
- 4) At each meeting, issues highlighted by the DSMB, the data-capture system or monitoring visits were largely discussed and whenever needed, consensus exercises were carried out.
- 5) In addition, regular monitoring visits always included an international gynaecologist oncologist with large colposcopy experience who evaluated the performance of the local colposcopist and reinforce procedures SOPs and virtually followed-up if improvement measures were adopted.
- 6) It is also worth to mention that the clinical management protocol of the ESTAMPA study was designed by international experts on colposcopy and cervical screening, and has been enriched over the years with input from the Data, Safety and Monitoring Board (DSMB) members, including suggestions to refine data collection on colposcopy, biopsy and treatment of

precancerous lesions. The DSMB also recommended that study colposcopists enrolled in the IFPCP/IARC colposcopy course, of which several ESTAMPA colposcopists are certified trainers. All ESTAMPA colposcopists, who are not trainers, have already enrolled/completed the IFPCP/IARC course.

4.- Figure 3 – visit 3. Why are the remains of the self-sample discarded and not stored for future potential biomarkers (an area which is quickly developing)? It seems strange that the clinician samples are stored but not the self-samples.

Response:

The use of self-sampling was added in a later version of the protocol as at the time of study design triage on self-sampling was considered not feasible. In fact, we decided to incorporate self-sampling in order to increase attendance to the 18 months follow-up visit. We decided to collect samples using a Copan swab to be washed on PreservCyt medium or to use a careBrush to be stored on STM whenever this was locally available. The 18 months visit is ongoing and self-sampling has not been used largely because about 88% of women recalled at 18 months have attended after simple reminder (by phone), and the remaining more after several calls. However, we take the point from the reviewer and we will discuss with our collaborators the possibility of following the same protocol as with clinician-collected samples at this visit (2 aliquots of 2 ml).

Minor comment

-

Last bullet point of strengths and limitations of this study. Would it be clearer if it said something such as “colposcopy and collection of biopsies was not performed in all negative women (only in a subset who were cytology positive), potentially

Response:

The clarity of this bullet point was also raised by the third reviewer, Dr Arbyn, therefore, we have decided to modify this potential study limitation, considering both comments and suggestions, as follows:

- *Colposcopy and collection of biopsies was not performed in HPV negative women (only in a subset who had abnormal cytology), potentially introducing verification bias when assessing absolute performance measures of screening tests to be used in primary screening; however, the study design will allow unbiased evaluation of triage tests*

Reviewer: 2

Reviewer Name: KimonChatzistamatiou

Institution and Country: Aristotle University of Thessaloniki, Thessaloniki, Greece

Please state any competing interests or state ‘None declared’: None declared

Please leave your comments for the authors below

Dear authors,

The submitted article describes the study protocol of the ESTAMPA study, multicentric study of cervical cancer screening with human papillomavirus testing and assessment of triage methods, conducted in Latin America.

The subject is very important since the advent of primary HPV screening, worldwide, makes the identification of accurate triage methods for the HPV positive women imperative. The study is well designed so that it can address the research hypotheses set by the investigators, and the sample size is adequate

e. It is also important that during the study the investigators will form a biobank useful for examining novel triage methods in the future.
The methods are described in detail and the limitations of the study are addressed.

Response:

We thank the reviewer for the positive comments.

Reviewer: 3

Reviewer Name: Marc Arbyn

Institution and Country: Sciensano, Brussels

Please state any competing interests or state 'None declared': None

Please leave your comments for the authors below

GENERAL COMMENTS

Given the high level of evidence that secondary prevention of cervical cancer using HPV-tests protects better than cytology, more and countries are switching to HPV-based screening. However, since the cross-sectional specificity of HPV testing is lower than cytology, appropriate triage of HPV-positive women is crucial.

Almonte et al engaged in an ambitious plan to investigate the accuracy of several triage markers nested in a large screening study set up in Latin America.

The ESTAMPA has the potential to become a landmark study which may trigger introduction of HPV screening in this continent and contribute to the WHO goal to make cervical cancer a rare disease;

The study recruits 50,000 women for screening among whom ~5000 women might be hrHPV+. If only ~5000 hrHPV+ women are used to address triage questions, the investment of enrolling such a large screening population is considerable. Finding ~5000 hrHPV+ women in an existing screening programme would look more efficient. It might be useful to add a hypothesis regarding the accuracy of the used screening tests (for instance CIN2/3+ detection rate ratios cyto vs hpv vs contesting. Another justification could be: given absence of HPV-based screening in Latin America, enrollment of a large screening population is needed in order to find sufficient hrHPV+ women where triage hypotheses could be assessed with sufficient power.

Response:

In effect, the study sample size has been set to evaluate multiple available and future novel triage tests and strategies combining several of them.

We agree that using an existing screening programme may be more efficient to evaluate the triage tests among 5000 women than rolling out such a large screening study. However, such a HPV-based organised cervical cancer screening programme does not exist yet in Latin America and is only being rolled out in a few European countries.

Mexico was the first Latin American country to introduce large-scale HPV testing in primary screening, however, the clinical management and capacity needed to guarantee the follow-up of HPV positives was not well assessed and the health system was not ready to perform a large number of colposcopies in the areas where HPV screening was first rolled out. The programme is now being reconsidered.

Argentina started the change from cytology-based to HPV-based screening in a different way, first in an area to investigate the barriers and facilitators of the process, the approaches and the logistics most suitable for successful implementation of screening. The programme goes on and is increasing the number of regions that is covered but it is a long process, and country leaders would have not allowed the somehow complex ESTAMPA study protocol to be carried out in women attending the new programme that still needed to be well established.

Similarly, El Salvador also started HPV-based cervical screening but followed by ablative treatment, and again the characteristics of the programme, particularly regarding the HPV-and-treat scheme would have not allowed the use of the ESTAMPA protocol.

This is why we decided to run the study in as many as possible countries in the region. In addition, each participating centre is being benefitted from extensive training of health providers and young researchers, of becoming partners with other colleagues around the region and most importantly for giving adequate screening to women who otherwise would have possibly never been screened.

The pages are not numbered, the line numbering is not adjusted to the line spacing and only the last digit of the line numbering is visible. This complicates the review process.

Response:
This has been corrected.

It would be much clearer for the reader if the CIN nomenclature should be applied as only terminology to describe severity of cervical lesions in biopsies. SIL could be reserved to describe cervical cytology. LAST is very confusing and always needs adjectives "cytological" or "histological" and even requires definition of the topography "cervical". Using the CIN and SIL nomenclature throughout the whole paper would increase the clarity and reduce the word counts.

Response:
We understand that the use of the LAST terminology can be somewhat confusing because it uses the same nomenclature as the Bethesda System developed for cytology. However, it is part of the study design from the beginning because we are interested in the evaluation of the performance of techniques to detect true precancer, excluding CIN2 where p16 is negative. We plan to continue to use it but it may be possible to revert to the CIN terminology.

I assume that the authors can easily address the given suggestions in a subsequent manuscript.

SPECIFIC COMMENTS

TITLE
No comments.

ABSTRACT
Introduction could be shorter

Response:
It has been shortened.

In spite of screening 50,000 women no objective regarding screening is mentioned. Only the triage objectives are mentioned.

Response:
We agree. We had several discussions about evaluating tests for primary screening. We decided to concentrate on triage testing for HPV positive women. Hence, our protocol did not include referral to colposcopy of HPV negative women (unless they had an abnormal Pap) and we thought our performance estimates of primary screening tests will be undoubtedly biased. However, we can definitely add this objective and estimate relative instead of absolute performance measures as suggested by the reviewer (see comment below). Thus, secondary objective number 1 (Page 7, lines 9-10) has been modified to:

"1) similar performance analyses among all recruited women and restricted to those with negative cytology"

And the sentence referring to evaluation of tests in primary screening in the Statistical Analysis Section (Page 17, lines 3-4) has been modified to:

“When evaluating tests in primary screening, relative measures of performance (e.g., relative sensitivity of HPV testing versus cytology or versus co-testing) will be used”

Methods and analysis

“with biopsy and treatment” “undergo colposcopy with biopsy and treatment”. Explain the conditions for biopsy and treatment”.

Response:

This is further explained in the study design section under “clinical management at first colposcopy”. Briefly, the clinical management will depend on the colposcopic impression (negative, positive minor, positive major, transformation zone type 3 cervix “TZ3”) and on the Pap results (HSIL or less than HSIL).

Among women with HSIL Pap results, those with a major colposcopy and those with a TZ3 are considered at very high risk of having cervical precancer or cancer. Therefore, those with major colposcopic impression will be treated with LLETZ and in those with a TZ3 an endocervical sample should be collected and an excision type 3 should be performed. Women with negative or minor colposcopic impression are considered to have “discordant results” which should be resolved at a multidisciplinary team (MDT) meeting. The result of this meeting could be to treat these women or to be recalled them 18 months later for a second HPV screen.

Among women with less than HSIL Pap results, biopsies should be collected whenever the colposcopic impression is positive (minor and major), an endocervical sample when a TZ3 is observed (and wait for the result to decide treatment).

Any woman with CIN2+ on biopsy should be treated, and any one with <CIN2 (no lesion or CIN1) as well as those that MDTs recommend will be recalled at 18 months for a second HPV, and attend colposcopy if HPV positive and again all CIN2+ will be treated.

The sample size could be mentioned (50 000 women screened, expected 500 CIN2+ cases).

Response: Numbers added.

STRENGTHS AND LIMITATIONS

4th bullet. Unclear since the ESTAMPA does not include an intervention to increase screening coverage.

Response:

As explained above, in Latin America, cytology-based screening has been offered opportunistically over decades and has not been effective in reducing cervical cancer. In many countries, health centres are given a target number of Pap smears to be done per year, and centres are focused on reaching such number independently of the age of women, whether they have had a smear very recently and furthermore, women with abnormal cytology are hardly ever adequately followed-up. In addition, recruitment areas have been purposely selected to offer screening to women whose access to health services is limited or very limited. This is why we believe that most women included in ESTAMPA would have not received proper screening otherwise.

In addition, different study centres have used specific approaches to invite women to screening: some centres have carried out a census of the area selected for the study, inviting one per one the women in the area door-to-door while doing the census (Paraguay and Honduras) that is getting a proper denominator for assessing participation, Other centres have used lists of women enrolled on the local health system attending services in the area selected (Costa Rica), the lists can be used as proxy denominators for assessing participation. In other place more opportunistic approaches have been used, particularly because of covering vulnerable populations, these approaches have included: i) collaborating with public health clinics to complete monthly quota of Pap smears while recruiting for ESTAMPA

(Colombia), ii) liaising with community leaders to create cancer awareness and invite women to specific health centres to be screened, among others.

In particular, in Paraguay different strategies to invite women were evaluated (number of phone calls, home visits, among others), a report on this is under preparation by the local team (Rodriguez-riveros MI et al, Implementation of strategies for the prevention of cervical cancer in women aged 30 to 64 years. Paraguay 2014-2018, ESTAMPA study, accepted for oral presentation at the 33rd International Papillomavirus Conference, Barcelona, Spain 2020).

All these experiences are contributing to later design strategies to increase coverage when scale-up implementation starts, focusing on those that have already in the local ESTAMPA small scale demonstrated high likelihood of success.

5th bullet. If the purpose of ESTAMPA would be to assess the absolute sensitivity and specificity, verification bias would be an issue. However for triage of hrHPV positive women, verification bias is not a problem. Even the assessment of the relative sensitivity of screening with cytology vs HPV vs cotesting would not really be affected by verification bias.

Response:

The clarity of this bullet point was also disputed by the first reviewer, Dr Castanon, therefore, we have decided to modify this potential study limitation, considering both comments and suggestions, as follows:

- Colposcopy and collection of biopsies was not performed in HPV negative women (only in a subset who had abnormal cytology), potentially introducing verification bias when assessing absolute performance measures of screening tests to be used in primary screening; however, the study design will allow unbiased evaluation of triage tests

INTRODUCTION

Line 2, ref 1. The following ref (PM:31812369) is more specifically describing the current burden of CC.

Response:

Reference added.

Line 12. Ref for effective and safe vaccines (PM:29740819).

Response:

Reference added.

Near end of 2nd § of introduction. Ref for increase uptake by offering self-sampling (PM:30518635).

Response:

Reference added.

MATERIALS AND METHODS

The study questions could be defined more concretely. A table could be added which defines the study's intermediate and final endpoints, with numerators and denominators.

Response:

The following table has been added to the manuscript.

Table 1. Definition of accuracy measures of screening tests.

Test for triage of HPV positive women		
	Numerator	Denominator

Sensitivity	No. HPV positive testing positive on triage with disease ¹	Primary: HSIL+ Secondary: CIN2+, CIN3+
Specificity	No. HPV positive testing negative on triage with no disease ²	Primary: <HSIL Secondary: <CIN2
PPV	No. HPV positive testing positive on triage with disease ¹	No. HPV positive testing positive on triage
NPV	No. HPV positive testing negative on triage with no disease ²	No. HPV positive testing negative on triage
Test for primary screening		
	Numerator	Denominator
Relative sensitivity (HPV/cytology)	No. HPV positive with disease ¹	No. abnormal cytology with disease ¹
Relative specificity (HPV/cytology)	No. HPV negative with no disease ²	No. cytology NILM with no disease ²
Relative sensitivity (HPV/co-testing)	No. HPV positive with disease ¹	No. HPV positive OR abnormal cytology with disease ¹
Relative specificity (HPV/co-testing)	No. HPV negative with no disease ²	No. HPV negative AND cytology NILM with no disease ²

HSIL: high-grade squamous intraepithelial lesion on histology. <HSIL: histologic diagnosis less than HSIL: negative, LSIL. HSIL+: HSIL or worse lesions.

CIN: cervical intraepithelial neoplasia. CIN2: CIN grade 2, CIN3: CIN3 grade 3. <CIN2: histologic diagnosis less than CIN2: negative, CIN1. CIN2+: CIN2 or worse lesions.

CIN3+: CIN3 or worse lesions.

NILM: Negative for intraepithelial lesions or malignancy.

¹ Women with disease: women with HSIL+ on review (Primary endpoint) or women with local CIN2+ or CIN3+ (Secondary endpoints).

² Women with no disease: women with negative, CIN1, LSIL histologic diagnosis, HPV negative women at 18 months and women with final (18 months) negative colposcopy.

Which HPV test will be used for screening?

Response:

The tests used for screening were selected based on: 1) being FDA approved for primary screening, b) being available or easy to deploy to the study centre/country. Six study centres used hybrid Capture 2 (HC2) and three used Cobas. In addition in one centre, under a local objective, 2311 samples were tested with HC2 and Aptima, and 991 with HC2 and Cobas. The results of the operational factors impact on test positivity have already been published (Robles C, Wiesner C, Martinez S et al, Gynecology Obstetrics, 2018)

What will be the conservation medium/media used for the screening/triage tests? PreservCyt only mentioned in the BIOBANKING §. This could be defined earlier.

Response:

PreservCyt medium is used for all samples collected, with the only exception when self-sampling has been used to increase attendance to the 18 months visit and the sample has been collected using a CareBrush QIAGEN preserved in STM.

This has been clarified in the manuscript. PC has been used to denote PreservCyt medium and has first been defined in Visit 1 – Methods Section (Page 12, line 21), , and later used whenever needed (Page 12, lines 22 and 35; Page 14 lines 22 and 24, Page 15, line 11).

The computation of the sample size could be better documented. It would be useful to add, in Supplemental materials, all elements included in the formula as well as the formula (with reference for the formula).

Response:

The sample size calculation follows the methodology proposed by Connor (*Connor RJ. Sample size for testing differences in proportions for the paired-sample design. Biometrics 1987;43(1):207-11*). The reference has been added to the manuscript and a supplementary file has been added explaining in detail the sample size calculation. Briefly, the sample size was computed for testing differences in proportions of sensitivity for HSIL+ of paired-sample triage tests according to equation (3) presented by Connor RJ (1987). We defined several differences in sensitivities, ranging from 5% to 10%, and combined with different levels of pairwise mismatch (ranging from 5% to 20%) between any two triage tests. In the supplementary file, we included some figures to show both the sample size and power at different parameters combinations.

Page with title "VISUAL METHODS on line 3"

Sentence on top: add "hrHPV and type-specific" before "persistence".

Response: "hrHPV and type-specific" added.

Third last sentence

HPV persistence as a triage strategy

Only 1-3 months is very short to assess persistence. Can this not be defined at 3-6M?

Response:

We agree with the reviewer, 1-3 months is too much a short period to assess HPV persistence, which will be assessed using the HPV test done at 18 months.

However, we are now exploring the possibility of using short-term persistence (1-3 months) as a substitute of colposcopy as a strategy to

Last §. Line 54. "Some of the methods already...". Replace by: "Some of the screening HPV tests already ...". Replace "individual" by "separate".

Response:

"Some of the methods already..." has been replaced by: "Some of the screening HPV tests already ..." and "individual" by "separate".

METHYLATION

Note the enormous heterogeneity in target genes (viral or human) and detection systems currently described.

Response:

We agree with the reviewer. We are aware of the large heterogeneity of target genes and detection systems. There is not a consensus which of these designs will produce the best classifiers going forward. However, it is known that methylation tests of both HPV and human genes (at least one human gene) provide synergistic information and some studies have shown that both genes may have a greater sensitivity for CIN3 detection. Additionally, a combined multitype methylation assay including late viral capsid genes, L1 and L2, in most important carcinogenic HPV types may serve as a triage test for HPV-positive women. We envisage to concentrate the most validated and promising techniques that can have high performance as triage.

PARTICIPANTS

Exclusion of patients with treated precancer (6 months). Period is quite short. The first post treatment examination often is at 6M. Consider a longer exclusion period: at 9 M.

Response:

We understand the concern, however, women have already been recruited following such exclusion criteria and we would not be able to change it.

Nevertheless, as explained before most of ESTAMPA participants were not properly screened before, and we expect very few of them (proportion not assessed yet) having been treated for cervical precancer previously. Furthermore, we may estimate the proportion of women in the study recently treated, although not precisely, by inspecting comments by colposcopists of particular features such as clear signs of recent treatment.

STUDY DESIGN

Line 3 referred to colposcopy with biopsy. Add “if needed (see below)”

Response:
“if needed” added.

The treatment visit is not mentioned. Also post-treatment follow-visit(s) is/are not mentioned

Response:

The study protocol establishes that once a woman is treated for cervical precancer, she exits the study. LLETZ is done at the colposcopy room and the colposcopic impression and other features at the time of LLETZ are recorded for further analysis.

Local health providers as well as the DSMB have had concerns regarding the study exit of women once they are treated. Study colposcopists who usually work on a hospital that covers the area selected for ESTAMPA, have committed to follow-up treated women.

It has been agreed that the first follow-up visit should be done at 6 or 12 months, in accordance with local regulations. The treatment follow-up visit can include an HPV test or a Pap smear followed by colposcopy if the test is positive.

This has been further explained in the manuscript in two sections; first under the Clinical management at first colposcopy Section (Page 14, lines 9-13):

“The colposcopic impression using a standard colposcopy nomenclature and specific colposcopy features (e.g., size and location of observed lesions, number and severity of biopsies collected) are recorded in colposcopy study forms.

At the treatment visit, first a colposcopy is done followed by LLETZ. The reason and type of LLETZ as well as the colposcopic impression are recorded in treatment study forms”.

And under the Visit 4: Final colposcopy Section (Page 15, lines 5-9):

“All women exiting the study are given a report with their screening and diagnosis results and clear indications on how to continue with routine follow-up care. Study colposcopists who usually work on a hospital that covers the area selected for ESTAMPA, have committed to follow-up treated women. It has been agreed that follow-up may be done by HPV testing or Pap with colposcopy of those HPV positive or with abnormal smears, and that the first follow-up visit should be done at 6 or 12 months in accordance with local regular care.”

VISIT 1

Line 32. Two consecutive brushes (which brushes?). Note that the endocervical brush (fi Cytobrush) is inappropriate for cytology (Cervical broom or combination with extended spatula) is recommended.

Response:

Cervex (Rovers Medical Devices) brushes are used for sample collection. The first Cervex is smeared into a glass slide ahead of being washed in PreservCyt medium, which will be used for HPV testing and LBC (only in centres where ThinPrep equipment is available). We had started an external review of Pap smears and so far reviewers have found that the Pap prepared slides have enough quality for interpretation. The second Cervex brush is washed in PresevCyt medium immediately after collection and later aliquoted for triage testing.

It would be better to use only one term for cytology “LBC” instead of Pap smear which usually means conventional cytology.

Response:

As explained above, we do perform conventional cytology. In addition and only in a few study centres, LBC preparations are also done.

Line 45 “those testing positive in any QC test become the study cohort”. If the women testing at QC (colpo visit), then the persistence cannot be assessed anymore. Consider not including the QC results in the study cohort or to include the women concerned only in a sensitivity analysis.

Response:

We feel ethically obliged to add HPV negative women who test positive in QC into the study cohort so that these women under proper diagnostic evaluation since they are HPV positive. However, we do agree with the reviewer on the potential of using their HPV results equivocally for HPV persistence evaluation. For such purpose and possibly for other triage testing evaluations, these women will be excluded from statistical analysis.

Page with CLINICAL MANAGEMENT AT FIRST COLPOSCOPY

Line 11. “to avoid inducing bias”. Replace by “to avoid breaching the principle of independence of the outcome assessment”

Response:

“to avoid inducing bias” has been replaced by “to avoid breaching the principle of independence of the outcome assessment”

After the VIA, the colposcopist inspects. I assume it will be another colposcopist than VIA inspector. In this case “another” should be added before colposcopist.

Response:

VIA is done by a nurse, midwife or a general doctor. Colposcopy is done by a “colposcopist”, a gynaecologist or gynaecology oncologist trained in colposcopy. The nurse, midwife or general doctor does not discuss his VIA diagnosis with the colposcopist, and viceversa.

VISIT 3;

Self-

collection. Hybridisation techniques such as cHPV and HC2 should be avoided since proven to be significantly less sensitive and specific for CIN2+ and CIN3+ on self-samples compared to on clinician-taken samples. Only validated PCR methods should be used on self-samples (Ref PM:30518635)

Response:

We fully agreed with the reviewer. However, self-sampling was introduced in a recent version of the protocol as a tool for local investigators to increase attendance to the 18 months follow-up visit. We decided to include either the use of HC2 sampling kits (CareBrush QIAGEN, STM) when they were locally available or the use of a dry swab to be washed in PreservCyt medium. As explained to the first reviewer, The 18 months visit is ongoing and self-sampling has not been used largely because about 88% of women recalled at 18 months have attended after simple reminder (by phone), and the remaining more after several calls.

Line 41. Cytobrush is insufficient for a representative cytology (see above).

Response:

The cytobrush, is the Cervex brush, which is washed on PreservCyt and is only used for HPV testing. We do not perform cytology at the 18 months follow-up visit.

PATHOLOGY

Last §. CIN will be used for histology classification. This is an excellent terminology. As remarked earlier, for reasons of consistency and clarity CIN should be used as only terminology throughout the paper.

Response:

We disagree with the reviewer. As explained above, we understand that the use of the LAST terminology can be somewhat confusing because it uses the same nomenclature as the Bethesda System developed for cytology. However, it is part of the study design from the beginning because we are interested in the evaluation of the performance of techniques to detect true precancer, excluding CIN2 where p16 is negative. In addition, the external histology review is already undergoing using as final adjudication endpoint HSIL+. Nevertheless, we will use as secondary endpoints: local CIN2+ as standard cutoff for

treatment and local CIN3+ as likely representing better “true cervical precancer”. Thus, we plan to continue using LAST it but it may be possible to revert to the CIN terminology for reporting.

Page with SAMPLE SIZE CALCULATION
See remark in General Comments (above).

Response:

Already answered above. A supplementary file with details on sample size calculation has been added.

For the assessment of the accuracy of triage of hrHPV+ strategies/markers, verification bias is not an issue. Cases with CIN2+ found only by cytology are not part of the Triage Cohort. (see above).

Response:

This has been noted. We agree with the reviewer and has edited the last bullet point of strengths and limitations accordingly.

Last sentence in first §. Challenges regarding participation. This sentence is very vague. Is there an indicator to assess participation? Follow adherence, treatment compliance could be easily assessed from the study design.

Response:

We agree. We can certainly evaluate adherence to the screening process, e.g., attendance to different visit, adherence to treatment, time delay between screening, colposcopy and treatment, etc. Regarding screening participation, as explained above, different study centres have used specific approaches to invite women to screening: some centres have carried out a census of the area selected for the study, inviting one per one the women in the area door-to-door while doing the census (Paraguay and Honduras) that is getting a proper denominator for assessing participation, Other centres have used lists of women enrolled on the local health system attending services in the area selected (Costa Rica), the lists can be used as proxy denominators for assessing participation. In other place more opportunistic approaches have been used, particularly because of covering vulnerable populations, these approaches have included: i) collaborating with public health clinics to complete monthly quota of Pap smears while recruiting for ESTAMPA (Colombia), ii) liaising with community leaders to create cancer awareness and invite women to specific health centres to be screened, among others. We will summarise the screening invitation approaches used in a separate HPV-based cervical screening report, and centres such as the Paraguayan one will report their uptake strategies separately (Rodriguez-riveros MI et al, Implementation of strategies for the prevention of cervical cancer in women aged 30 to 64 years. Paraguay 2014-2018, ESTAMPA study, accepted for oral presentation at the 33rd International Papillomavirus Conference, Barcelona, Spain 2020).

DISCUSSION

Last page.

Line 4 “could mask final results”. Better; “could influence results”.

Response:

“could mask final results” changed to “could influence results”

How was multiple testing and inter-centre heterogeneity included in the sample size computation. Can be detailed in Suppl methods (see above).

Response:

This is explained in the supplementary file with details on sample size calculation. Briefly, using Bonferroni correction we adjusted the type I error for multiple comparisons (up to 10) and show that statistical power will remain above 80% to detect 5% differences in sensitivities for tests with pairwise discordance lower than 8%. Inter-centre heterogeneity was

not included in the power calculation but we will adjust the analyses by study centres and will adopt measures that allow controlling for this feature.